# Integrated UAV photogrammetry and SMRM constitutive modeling for spatial stability assessment of layered rock slopes

Huagang Shan[1], Zhongliang Wang[2], Jie Wu[3]*, Weiguo Wang[1], Jianyang Lu[4], Faquan Wu[2], Zhenghua Li[1]

1 Shaoxing Kezhu Expressway Co., Ltd., Shaoxing, China, 2 Facuty of Civil Engineering, Shaoxing University, Shaoxing, China, 3 Zhejiang Rock Innovation Technology Co., Ltd., Shaoxing, China, 4 Zhejiang Digital Transport Institute Technology Co., Ltd., Shaoxing, China

* 328016787@qq.com

## Abstract

Layered rock slopes with structural discontinuities pose significant challenges for stability assessment under complex geological conditions. This study presents an integrated approach combining unmanned aerial vehicle (UAV) photogrammetry, 3D numerical modeling, and the Statistical Mechanics of Rock Masses (SMRM) constitutive model to evaluate slope stability. Focusing on a layered tuffaceous sandstone slope in Shaoxing, China, UAV-derived point cloud data were processed via Cloud-Compare and Rhino software to generate a high-resolution 3D geological model. The CC-Griddle-Flac3D workflow enabled efficient meshing and integration with Flac3D for numerical analysis. The SMRM model was implemented to quantify anisotropic deformation, stress distribution, rock mass quality, and failure probability. Results demonstrated that the proposed modeling framework effectively captured slope deformation trends and stress heterogeneity, The direction of the structural plane is highly consistent (24 ° simulated vs. 23 ° observed), which indicates that the SMRM constitutive model shows high accuracy in characterizing the anisotropic properties of materials. The SMRM-based stability metrics revealed localized weak zones (Grade IV rock mass) with elevated reinforcement demands ($\Delta\sigma_3 = 0.0126$ MPa) and failure probabilities (60%), while maintaining an overall stability coefficient of 2.0. This method enhances the spatial visualization of rock mass parameters, enabling targeted reinforcement strategies. The study provides a validated technical pathway for rapid slope stability evaluation in geologically complex regions, supporting data-driven disaster prevention decisions.

## 1. Introduction

Layered rock slopes are a type of geological structure formed when the dip of rock layers aligns with the slope dip, making them prone to deformation and failure. Such

**Data availability statement:** S1 File. Information about rock material parameters for slope stability analysis. 10.6084/m9.figshare.29377223 (XLSX) S2 File. Information about Statistical window structure plane parameter table. DOI:10.6084/m9.figshare.29377235 (XLSX).

**Funding:** Scientific Research Project of Zhejiang Provincial Department of Transportation; 2024GCKY01. The Central Guidance on Local Science and Technology Development Fund of Projects (2024ZY01041).

**Competing interests:** The authors have declared that no competing interests exist.

slopes are vulnerable to landslides and failures under external influences, such as rainfall or human activities, leading to significant economic losses or even casualties [1]. Rock slopes typically experience large-scale damage, small deformation, and strong randomness in instability [2], making their stability prediction and analysis extremely challenging. The instability of rock slopes, especially in regions prone to geological disasters, is a crucial issue and has been a focus of research in geotechnical engineering.

Many scholars have conducted a series of studies on the stability of layered rock slopes. Wei Sun [3], for instance, used friction tests and numerical simulations to compare the failure mechanisms of layered rock slopes under different dip conditions, providing references for the safety evaluation and disaster recognition of slopes with inclined layers. Haibing Yu [4] conducted in-depth research on the dynamic stability evolution mechanism of layered rock slopes under rainfall infiltration through a combined numerical simulation and theoretical analysis approach. Ruoting Liu [5] analyzed the slope deformation response process and basic failure patterns under different working conditions at the Jiniou Hydroelectric Station, identifying its main triggering factors and formation mechanisms. Zhengzheng Cao [6] systematically investigated the overburden fracture evolution and surface crack development characteristics during the initial mining stage using numerical simulation techniques.

With the development of three-dimensional rock mass technology, 3D modeling and numerical simulation techniques are increasingly used to better capture and express the geological characteristics of layered rock slopes [7]. As a result, rock mass 3D technology has become widely applied as an efficient method in geotechnical engineering analysis. In 1986, ITASCA Corporation in the United States developed Flac (Fast Lagrangian Analysis of Continua), widely used in geotechnical, tunnel, and mining engineering research [8,9]. A limitation of Flac3D is the difficulty in pre-processing for modeling, for which ITASCA introduced the Griddle mesh processing plugin to achieve meshing of 3D models.

In recent years, many scholars have made innovations in complex 3D geological models. Bin Hu et al. [10] used Fortran programming to establish Flac3D models quickly and accurately for complex terrains. Xiaolong Deng [11] combined 3D laser scanning data and Hypermesh software to efficiently establish numerical models for Flac3D with high precision, achieving the "visual" and "computable" goals. Jiangtao Tang et al.. Bock S [12] is based on the FLAC3D code 's ability to import grids using the IMPGRID command of a third-party grid generator. [13] wrote a VB interface program between Flac3D and ANSYS to rapidly and intuitively create Flac3D models. Jinlong Chen et al. SONG Weihua [14] proposed a ground stress inversion method combining rhinoceros fine modeling and transverse isotropy theory for the geological structure area, and carried out experiments near the Guodishan fault. LI Chenguang [15] established a three-dimensional numerical model of Guanyinshan tunnel portal section by using Rhino, AutoCAD and FLAC three-dimensional software, and simulated the whole process of tunnel portal construction.

Xiujun Liu [16] used GOCAD's modeling capabilities and the Flac3D interface program to rapidly establish complex geological models. Wenjie Xu et al. [17] used

ADINA software, known for its excellent mesh division capabilities and interface with CAD/CAM software, to convert models into formats compatible with Flac3D, addressing the limitations of Flac3D's pre-processing modeling. These technological advancements have provided strong support for slope stability research.

The visualization of 3D rock masses is a new research direction in the field of geological engineering. With the development of modern information technology, 3D modeling and simulation analysis of rock masses have gradually become essential tools in engineering applications [18]. Traditional rock mass analysis methods often rely on field observation and testing, but the complexity and heterogeneity of rock mass structures make it difficult to fully capture the detailed features in the field [19]. Therefore, how to accurately express the structural characteristics and mechanical behavior of rock masses using 3D technology has become an urgent research problem.

To further study the development trend of slopes and rapidly assess emergency impacts, this paper takes the layered rock slopes in Shaoxing, Zhejiang, as an example. Based on drone photogrammetry and CC-Griddle-Flac3D joint modeling, drone technology is used to quickly obtain terrain and geological data, and complex 3D geological models are created using the Rhino platform and Griddle plugin. By combining the SMRM constitutive model and Flac3D, this study provides a new technical approach for slope stability analysis under complex geological conditions. The study visualizes the rock mass's parameter states and spatial distribution, quickly assesses the development trend of slopes during excavation, and provides references for emergency decision-making in related engineering projects.

## 2. Engineering overview and geological conditions

### 2.1. Engineering overview

The study area is the Kezhu Expressway, located in the central-western part of Shaoxing City, Zhejiang Province, China. The route starts from the central part of Keqiao and extends southwest to the northern part of Zhuji City, with a total length of approximately 39.262 kilometers. Among this, the Keqiao section is about 10.662 kilometers long, and the Zhuji section is about 28.600 kilometers. The design speed is 100 km/h, and the geographic location is shown in Fig 1. The Kezhu Expressway is an important part of the G9903 Hangzhou Metropolitan Area Ring Expressway and has been included in the national expressway network plan.

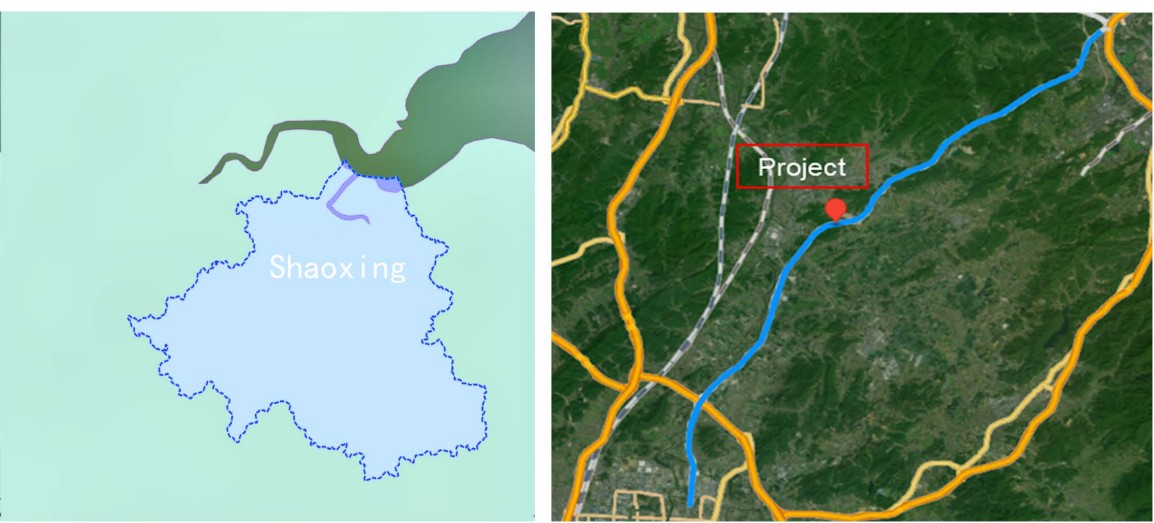

**Fig 1. Geographic location of the study area Map generated using Natural Earth data, No copyright restrictions apply.**

No specific permits were required for this study because it did not involve human participants, animal subjects, endangered or protected species, or access to protected areas. All data collected and analyzed were from publicly available sources.

Along the Kezhu Expressway, there are 22 locations of high-cut slopes, over 40% of which are layered rock slopes. The primary lithology is tuff and tuffaceous sandstone, often interspersed with weak interlayers. The volcanic detritus strata have complex rock mass structures and highly variable physical and mechanical properties, making them prone to landslides under external factors, which can cause significant economic losses and even casualties. Additionally, slope instability has a strong tendency to occur suddenly. Therefore, to ensure the safety of the construction project and workers, effective reinforcement measures must be taken promptly. To achieve this, it is necessary to conduct in-depth studies on the deformation patterns, reinforcement requirements, and stability coefficients of the slopes, and propose reasonable construction methods and reinforcement plans based on these findings.

## 2.2. Geological conditions

The Kezhu Expressway is located in the central-western part of Shaoxing City, Zhejiang Province, as shown in Fig 2(a). The route generally follows a topography with lower ends and a higher middle section. The region has intense tectonic activity, mainly influenced by the Huaxia, Huaxia-style, and New Huaxia tectonic systems, forming the geomorphic features of the Kuaiji Mountain range with a northeast-southwest orientation. The study area is a slope-hill accumulation valley region, where the thickness of the cover layer is generally 1–2 meters, with a thicker cover layer at the foot of the slope and silty clay containing gravel at the top of the slope. The bottom of the slope mostly consists of fill soil or strongly weathered tuffaceous sandstone. Due to long-term tectonic activity, the rock mass is intensely weathered, and some areas are sand-like in texture. Layered phenomena along the slope are commonly observed, with dip angles between 25° and 60°, as shown in Fig 2(b).

The geological section is composed of strongly to moderately weathered tuff and tuffaceous sandstone. The engineering geological cross-section is shown in Fig 2(c). The strongly weathered rock masses are gray-yellow, purple-red, or gray-green in color, with well-developed joints and fractures, making the rock mass fragmented and classified as soft rock. The moderately weathered rock masses are purple-red and gray-green, with blocky structures and layered formations, often containing mudstone or fine sandstone interlayers. These rock masses are more fragmented, harder, and have a brittle hammering sound, classified as sub-strong rocks. The major joints cut through the rock mass, forming wedge-shaped blocky structures. After slope excavation, rock blocks are prone to falling, and the structural joints are similar to the slope orientation, making the layered rock slopes more susceptible to instability, thus negatively impacting slope stability.

## 3. Slope surface morphology reconstruction based on point cloud

Flac3D, as a commonly used three-dimensional finite difference software, is increasingly applied in the engineering field. This study uses Cloud Compare (CC) and Rhino software for modeling processing. Through the "CC-Griddle-Flac3D" joint processing workflow, (Fig 3). the point cloud data is transformed into a more realistic three-dimensional geological model of the slope. The specific process is as follows:

(1) Determining the slope area: The slope location is shown in Fig 4(a). A DJI Mavic 3E drone was used, with the camera angle set to 90°and the flight path overlap rate set to 80%, with a lateral overlap of 60%. The flight path was planned in a parallel line pattern at a constant height to ensure a uniform flight altitude, making the process efficient and easy to operate.

(2) Generating the three-dimensional model from drone photogrammetry data: The photos taken during the flight were processed using point cloud generation software to create a 3D model in las format. The model was then converted into STL format for further use. (Fig 5(a)).

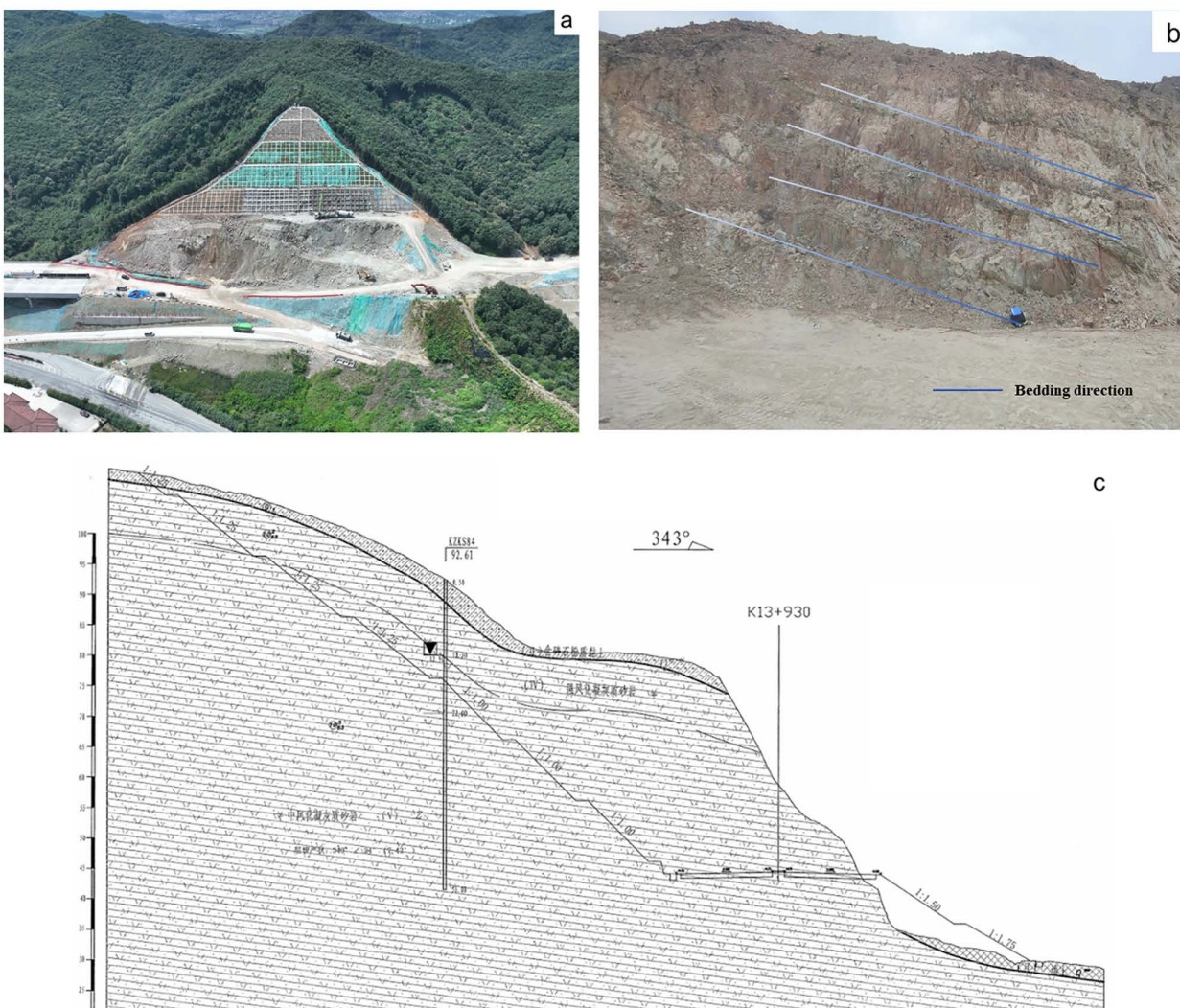

**Fig 2. Images of the study area. (a)** Aerial view of the study area **(b)** Photo of layered rock survey **(c)** Engineering geological cross-section (Original photograph taken by the authors during fieldwork at the Kezhu Expressway. Copyright held by the authors, released under the Creative Commons Attribution License (CC BY 4.0).).

(3) Point cloud decimation and meshing: The point cloud was imported into Cloud Compare, and noise was removed using the "Noise filter" function. The "Subsample" function was used for downsampling, completing the point cloud decimation (Fig 5(b)) to improve the efficiency. The Cloth Simulation Filter (CSF) method was used to remove vegetation data and retain ground information. This process separated vegetation points from ground points, ensuring that the numerical calculations considered only terrain features.The decimation point cloud was then meshed into a triangular grid (Fig 5(c)).

(4) Generating the three-dimensional geological solid model: The mesh data was imported into Rhino software. The modeling method of "mesh-surface-solid" was applied (Fig 5(d)-(f)). The Rhinoresurf plugin was used for surface reconstruction, and generate a three-dimensional solid model.

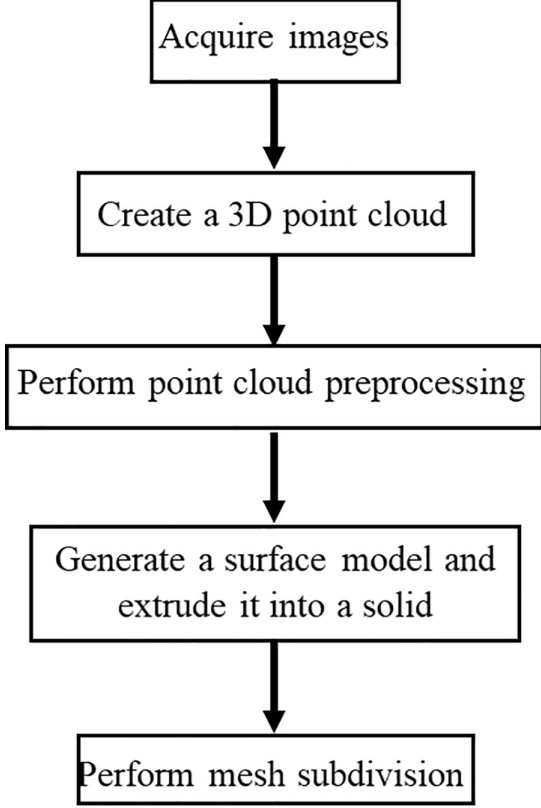

**Fig 3. "CC-Griddle-Flac3D" joint modeling process images.**

Generation of three-dimensional geological numerical model. In order to carry out numerical simulation analysis of the slope, it is also necessary to mesh the geometric model. This work is mainly done using the Griddl plug-in. The steps are as follows:

1) divide the initial grid by the ' mesh ' command.

2) on the basis of the initial grid, use the surface of the Griddle plug-in to re-divide the initial grid. The unit size of the slope is set to 8m, and the other parameters are: Mode: Tri, Minimum edge length: 20, Maximum edge length: 20, Ridge angle: 20 (Fig 5 (g)).

3) Use the volume function to convert the surface mesh model of Step 2) into a volume mesh model that meets the requirements of the flac3d mesh, and export it to a mesh file *.f3gird.that conforms to the flac3d format (Fig 5 (h)).

During the UAV aerial photography process, a total of 314 images were captured. These images were processed using Context Capture software to generate a 3D point cloud model (Fig 5(a)), yielding 252,797,102 three-dimensional point cloud data points. To address the modeling efficiency issues caused by the massive volume of raw 3D point cloud data, data lightweighting while preserving terrain features was achieved through point cloud preprocessing. Peripheral redundant points and noise were removed from the 3D point cloud model of the study area. An octree-based algorithm was applied to homogenize the point cloud, compressing the dataset to 4,027 points (0.0016% of the original data). The downsampled point cloud was triangulated and subjected to mesh smoothing, resulting in 8,021 mesh faces (Fig 5(c)).

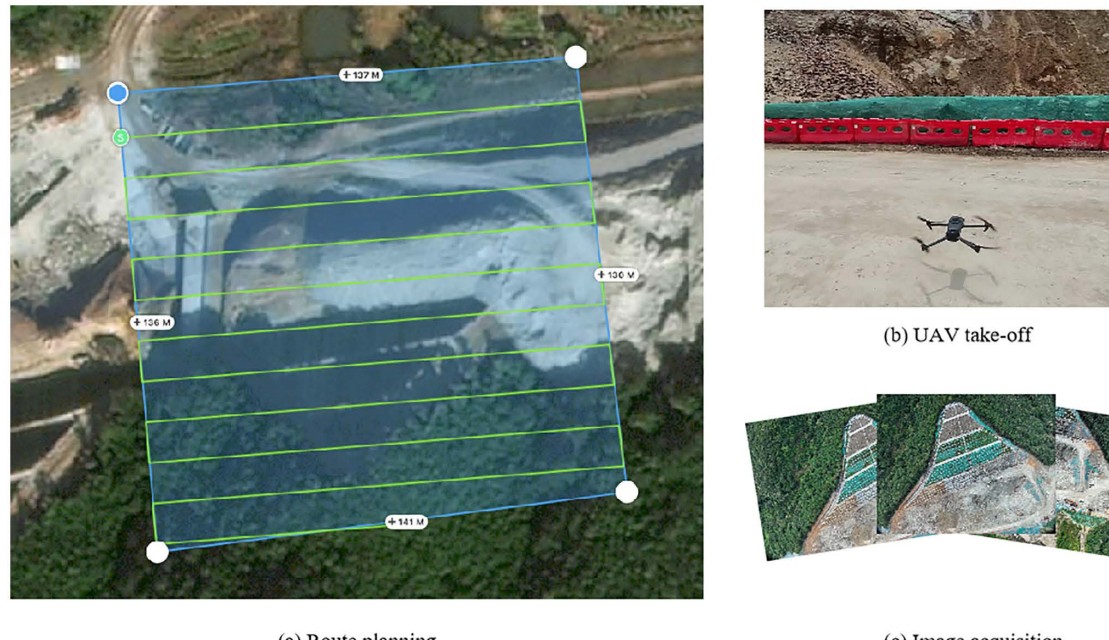

(a) Route planning

(b) UAV take-off

(c) Image acquisition

**Fig 4. Drone flight preparation and on-site work.** Original photograph taken by the authors during fieldwork at the Kezhu Expressway. Copyright held by the authors, released under the Creative Commons Attribution License (CC BY 4.0).).

After preprocessing, the data volume was reduced by 99.98%, while the key terrain features remained intact compared to the original point cloud. This optimized dataset provides a solid foundation for subsequent FLAC3D numerical simulation and analysis.

## 4. Constitutive model of statistical rock mechanics

### 4.1. Traditional constitutive model of FLAC3D

FLAC3D is a widely used numerical simulation software in the fields of geotechnical engineering and geological mechanics, providing essential tools for analyzing the deformation and failure of rock and soil bodies [8,9]. The Mohr-Coulomb constitutive model [20,21] and the ubiquitous-joint model [22] are two important traditional models in FLAC3D, applicable for simulating the mechanical behavior of homogeneous materials and materials containing joints, respectively.

The Mohr-Coulomb constitutive model is a classical model for describing shear failure in rock and soil bodies. Based on the Mohr-Coulomb yield criterion, it assumes the relationship between shear stress and normal stress as follows:

$$\tau = c + \sigma\tan\phi \tag{1}$$

Where $\tau$ is the shear stress, $\sigma$ is the normal stress, $c$ is the cohesion, and $\phi$ is the internal friction angle. The Mohr-Coulomb model is mainly used to describe homogeneous, isotropic materials (such as clay, sand, or intact rock). Its model parameters include elastic modulus, Poisson's ratio, cohesion, and internal friction angle.

The advantage of this model lies in its simple formula and wide applicability, effectively simulating elastic-plastic deformation and overall failure behavior of materials. However, its limitation is that it cannot reflect the anisotropic properties of rock and soil bodies, especially those with joints or weak surfaces, making it somewhat limited in complex geological conditions.

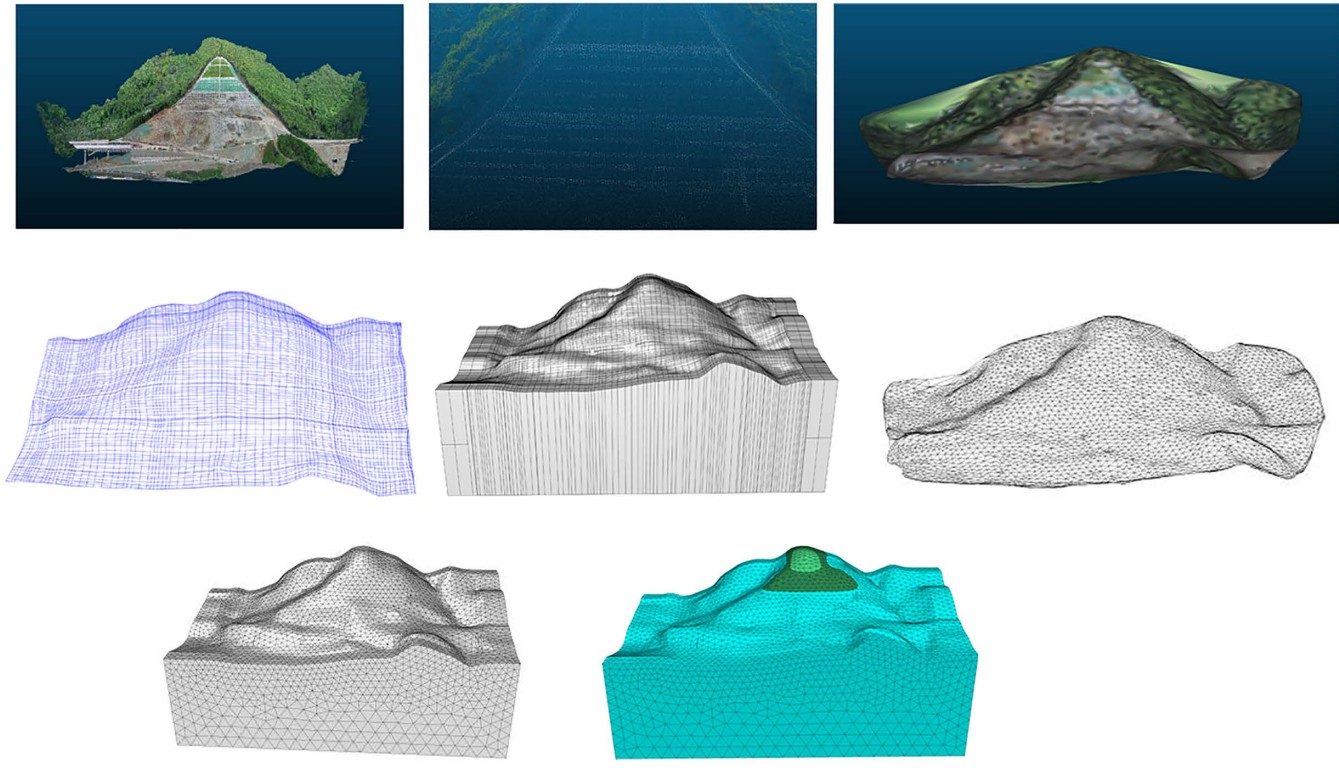

**Fig 5. "CC-Griddle-Flac3d" modeling.** Original model generated by the authors. Copyright held by the authors, released under the Creative Commons Attribution License (CC BY 4.0).

The ubiquitous-joint model is an improved constitutive model specifically designed to describe the anisotropic mechanical behavior of materials containing weak surfaces. This model adds joint parameters to the traditional Mohr-Coulomb model, allowing the simulation of the mechanical response of weak planes or joints commonly found in materials. Parameters such as joint direction (dip angle and dip direction), joint normal stiffness, and shear stiffness are introduced to describe the normal deformation, shear slip, and strength degradation characteristics of joints.

The ubiquitous-joint model achieves coupling analysis between the structural surfaces and the matrix by embedding the joint surfaces into the homogeneous material. This allows for effective simulation of deformation and failure controlled by joints in specific directions. Compared with the Mohr-Coulomb model, the ubiquitous-joint model is more suitable for simulating rock bodies with layering, jointed fractures, or other weak surfaces and is widely used in landslide analysis, slope stability studies, and jointed rock body tunnel design.

### 4.2. Improvement of the SMRM constitutive model

After decades of exploration and practice, numerical computation and simulation technology have become indispensable tools for analyzing geotechnical engineering problems. Statistical Rock Mechanics (SMRM) has been developing its own numerical calculation constitutive module, namely the SMRM constitutive model. Since numerical calculations have strict requirements for plasticity mechanics models, the stress-strain relationship in statistical rock mechanics itself includes the elastic-plastic deformation of the rock mass, thus the module described here should be capable of performing elastic-plastic deformation calculations.

The SMRM constitutive model is based on the elastic stress-strain relationship of statistical rock mechanics [23,24].The SMRM constitutive model combines the geometric probability model of rock mass structure, the energy principle of fracture mechanics and the continuum mechanics method. According to the energy superposition principle and the weakest ring theory, the constitutive model of jointed rock mass is established by considering the mechanical response of rock mass and structural plane.

The SMRM constitutive relation is given by Equation 2. On the basis of traditional numerical analysis for calculating stress, displacement, and other functions, this module adds extended calculations for commonly used geotechnical engineering indicators, including rock mass deformation modulus, SMRM rock mass quality score, failure probability of rock mass elements, and stability coefficient of rock mass elements. The extended calculations can output value contour maps for each indicator. This module leverages the advantages of numerical computation in expressing spatial patterns, displaying the spatial variation characteristics of these indicators, which helps grasp macroscopic patterns in engineering analysis.

$$\begin{cases} e_{ij} = e_{0ij} + \frac{\alpha}{E} \sum_{p=1}^{m} \lambda \bar{a}(k^2\sigma + \beta h^2 t_1)n_i n_j \\ \sigma_{ij} = \min(T_i\sigma_3 + R_i), \quad i = 1, 2...m \end{cases} \tag{2}$$

where i represents the rock block; m represents the structural surface group number; $e_{0ij}$ is the strain of the rock in the rock mass element; $\alpha$ is the dip angle of the structural surface; $E$ is the rock's elastic modulus; $\lambda$ is the structural surface density; $\bar{a}$ is the average radius of the structural surface; $k$ represents the normal stress state coefficient on the structural surface; $\sigma$ is the normal tensile stress; $\beta$ is the dip angle of the structural surface; h is the residual shear stress ratio coefficient; $t$ is the shear stress on the structural surface; $n_i, n_j$ are the cosine of the structural surface's normal direction; and $T_i$ and $R_i$ are the Coulomb strength coefficients.

Since the main program can call the triaxial principal stresses and their direction cosines from the loop step elements, the elastic modulus of the rock mass in the direction of the maximum principal stress can be conveniently obtained from Equation 3.

$$E_m = \frac{E}{1 + \alpha \sum_{p=1}^{m} \lambda \bar{a}[(k^2 - \beta h^2)n_1^4 + \beta h^2 n_1^2]} \tag{3}$$

where $E_m$ is is the elastic modulus of the rock mass.

The basic idea of the SMRM grading method is to calculate the deformation modulus of the rock mass in all spatial directions using statistical rock mechanics theory and then use empirical formulas to compute the corresponding SMRM score in each direction, thereby determining the rock mass quality grade in all spatial directions. Consistent with human thinking, the SMRM grading method uses a 100-point system with 5 grading levels. Using the RMR~Em empirical relationship proposed by Serafim and Pereira (1983) [25], the calculated rock mass deformation modulus Em in any spatial direction can be converted into an SMRM score on a 100-point scale.

$$\begin{cases} SMRM = \frac{1}{2}(E_m + 100) \quad E_m > 10GPa \\ SMRM = 40\lg E_m + 10 \quad E_m < 10GPa \end{cases} \tag{4}$$

According to the SMRM value, the rock mass quality is divided into five grades, as shown in Table 1:

**Table 1. Criteria for the classification of rock mass quality grades.**

| SMRM (RMR) | 81-100 | 61-80 | 41-60 | 21-40 | <21 |
|---|---|---|---|---|---|
| Rock mass grade | I | II | III | IV | V |
| Evaluation result | Very Good | Good | Average | Poor | Very Poor |

According to the theory of weak rings, if the rock or any group of structural surfaces in a rock mass unit fails, the rock mass unit is considered to have failed. Based on reliability theory, the failure probability of a rock mass unit is given by:

$$P = 1 - (1 - P_b)(1 - P_c)$$

(5)

where $P_b$ is the failure probability of the rock mass unit, and $P_c$ is the failure probability of the surface system.

According to statistical rock mechanics, rock mass reinforcement should involve limited artificial measures to stimulate and enhance the rock mass's self-stabilization potential, achieving effective control of rock mass deformation and stability. The reinforcement demand is defined as the minimum difference between the principal stress required to achieve ultimate stability and the actual principal stress:

$$\Delta\sigma_3 = \sigma_{3c} - \sigma_3$$

(6)

Statistical rock mechanics also provides the stability coefficient K at any point in the rock mass, defined as:

$$K = \frac{\tau_c}{\tau} = \frac{\sigma_1 - \sigma_{3c}}{\sigma_1 - \sigma_3}$$

(7)

where $\sigma_{3c}$ is the maximum shear strength, and $\sigma_1$ and $\sigma_3$ are the maximum and Minimum principal stress, respectively.

### 4.3. SMRM constitutive advantage

Based on the theory of statistical rock mechanics (SMRM), the SMRM constitutive model developed by Flac3D platform is developed and improved in this study, which effectively enhances its application ability in numerical simulation of rock mass engineering. Compared with the traditional model, the constitutive advantages of SMRM are as follows:

(1) The existing numerical calculation models of rock mass engineering are mainly based on continuous and isotropic media, and there are some limitations in the mechanical analysis of anisotropic media. Based on the geometric probability of rock mass structure and the energy superposition principle of fracture mechanics, the SMRM constitutive model superimposes the strain energy of rock mass under external force with the strain energy of structural plane network, and establishes a set of analysis theory for describing the mechanical behavior of equivalent continuous medium of rock mass.

(2) Based on the advantages of Flac3D in spatial distribution, the SMRM constitutive model can directly show the spatial variation characteristics of rock mass related indexes, and provide scientific basis for grasping the macroscopic law in engineering analysis. The anisotropy of rock mass parameters can be characterized by means of stereographic projection and anisotropy coefficient. The non-uniformity can be intuitively reflected by the point cloud map of the spatial position of the specific engineering rock mass. For engineering problems, the stress field distribution of the model can be obtained by numerical calculation of the SMRM model, and based on this, the spatial cloud diagram of key parameters including rock mass elastic modulus, SMRM rock mass quality index, rock mass unit failure probability, rock mass reinforcement demand and stability coefficient can be generated. As shown in Figs 6-10.

(3) The layered structural plane in the SMRM constitutive model is established in an 'implicit ' way. In the modeling process, the bedding structural plane is not considered first, and the corresponding bedding plane stress distribution can be automatically generated in the calculation by inputting the inclination and inclination of the bedding plane through the command flow. This method simplifies the modeling process and improves the modeling efficiency.

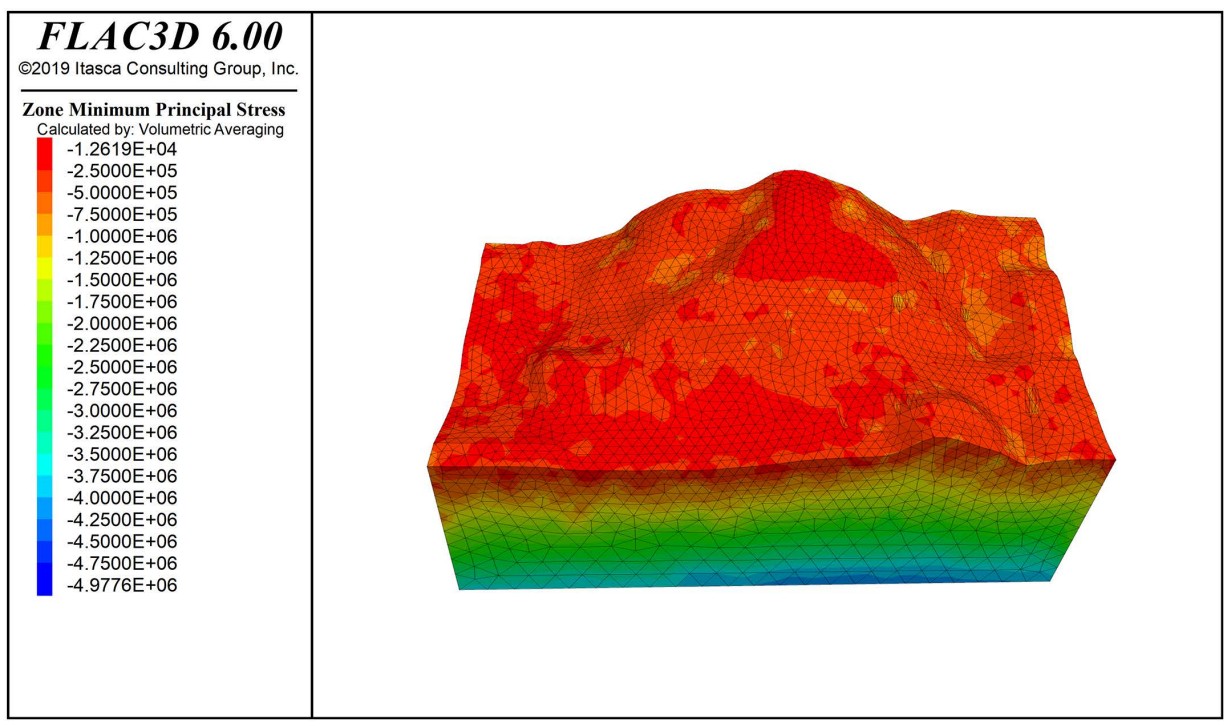

**Fig 6. Principal stress contour.**

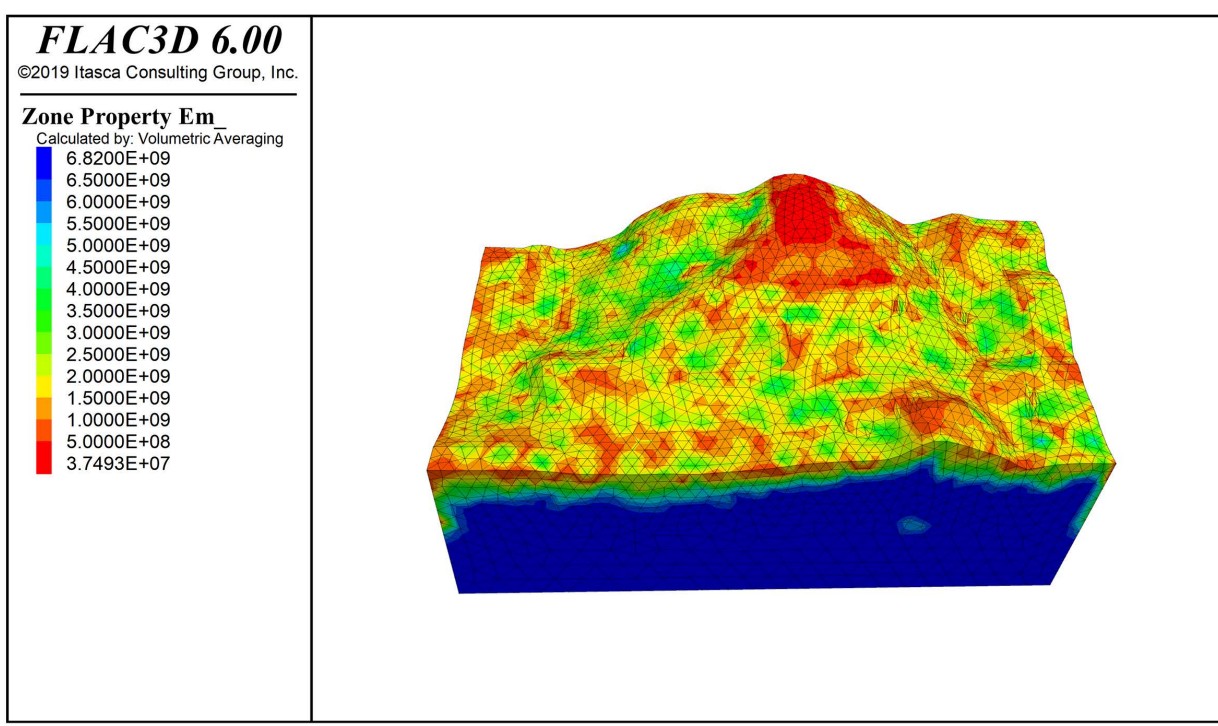

**Fig 7. Modulus of elasticity contour.**

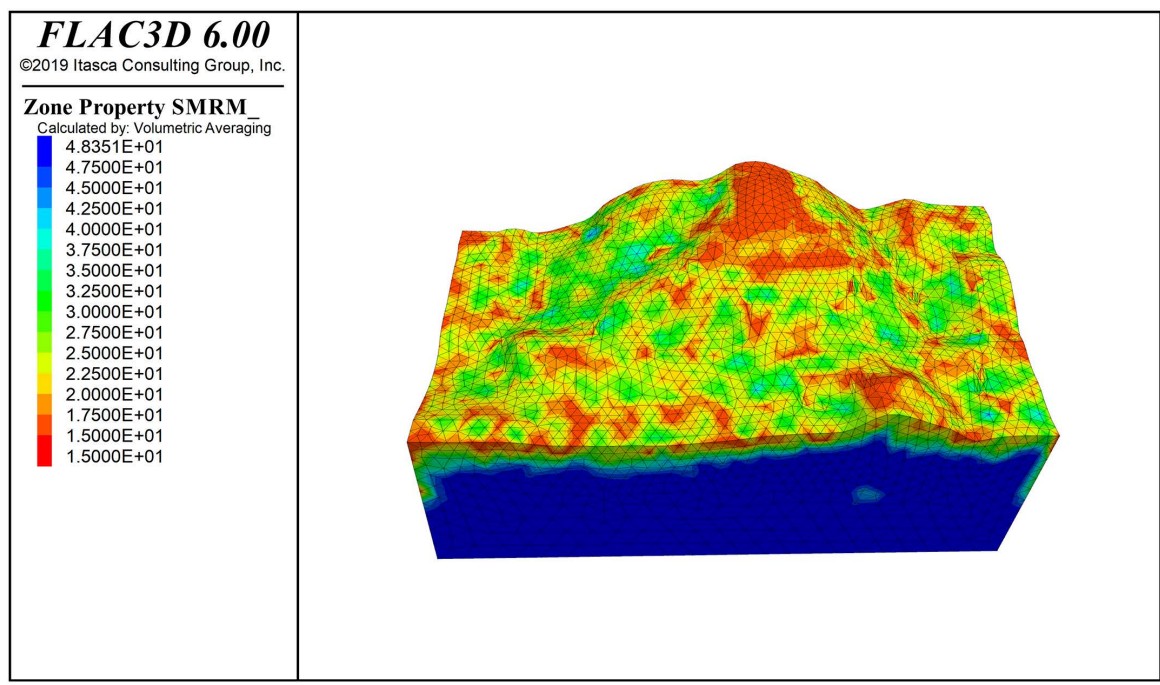

**Fig 8. SMRM rock mass contour.**

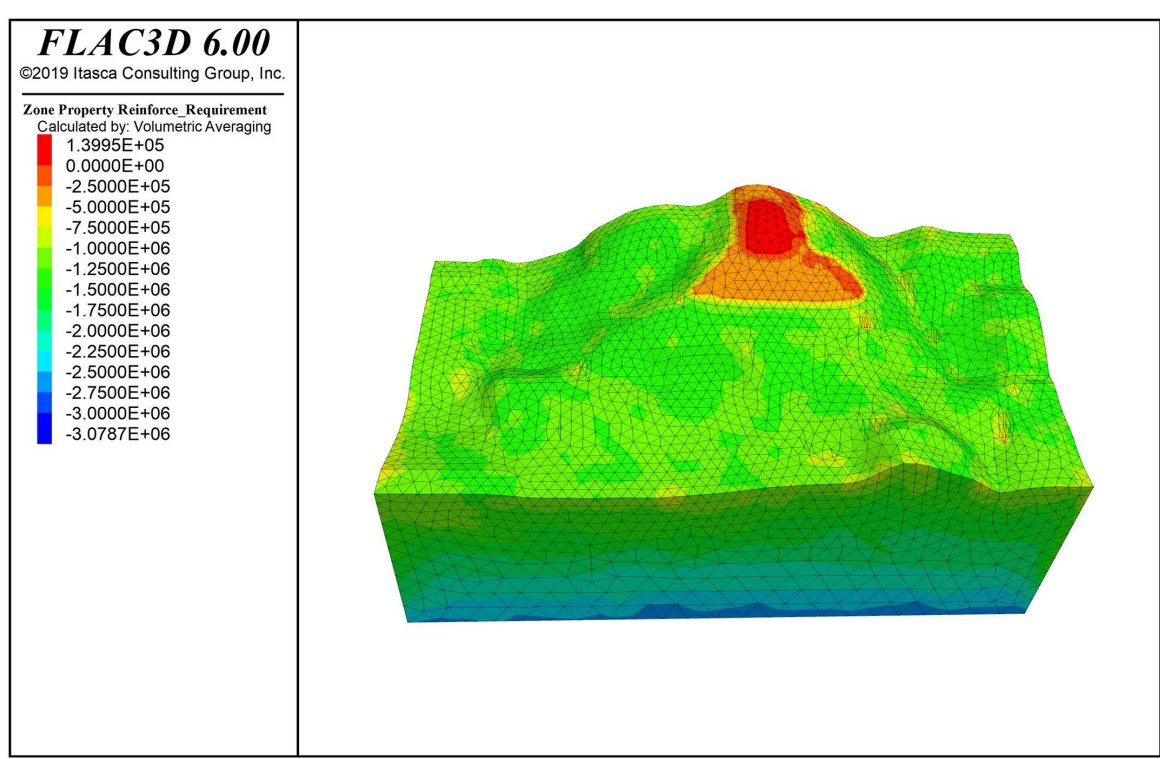

**Fig 9. Reinforce_Requirement contour.**

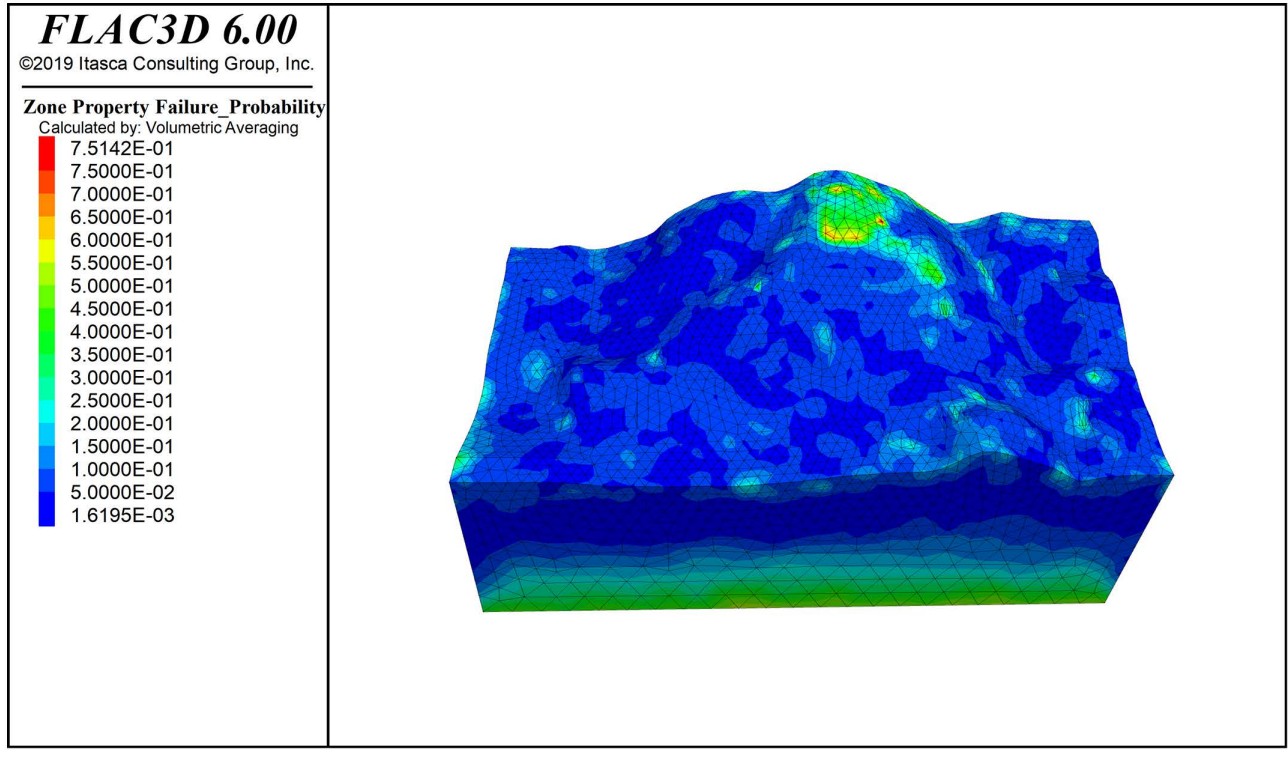

**Fig 10. Failure_Probability contour.**

## 5. Slope stability analysis based on the SMRM constitutive model

The SMRM constitutive model can help visualize the parameter state and parameter field of the rock mass in FLAC3D. In the subsequent extended calculations, it reflects both the directional variation characteristics of the rock mass parameters and their spatial variations. The heterogeneity of the rock mass can be reflected by the point cloud map of the spatial location of the engineering rock mass.

### 5.1. Three-dimensional model of layered slopes and parameter selection

The model uses the numerical model established in the previous chapter based on point cloud data, as shown in Fig 11. The positive direction of the X-axis points east, the positive direction of the Y-axis points north, and the positive direction of the Z-axis is vertically upward. The slope faces 340°, with the model calculation range being 440 meters horizontally, 260 meters vertically, and 200 meters in height. The numerical model simulates formations such as moderately weathered tuffaceous sandstone, strongly weathered tuffaceous sandstone, and sand-gravel silt clay. The model is subdivided into 32,626 units with 13,516 nodes.

Considering the terrain and the load direction of the slope, boundary conditions for the model are set with normal support constraints on the lateral (X-direction) and longitudinal (Y-direction) boundaries, three-dimensional constraints on the bottom boundary (Z-direction), and a free boundary at the top.

The rock mechanics parameters for the slope rock were selected based on the engineering geological manual and related engineering experience, combined with the slope model. The physical and mechanical parameters of the slope rock are listed in Table 2.

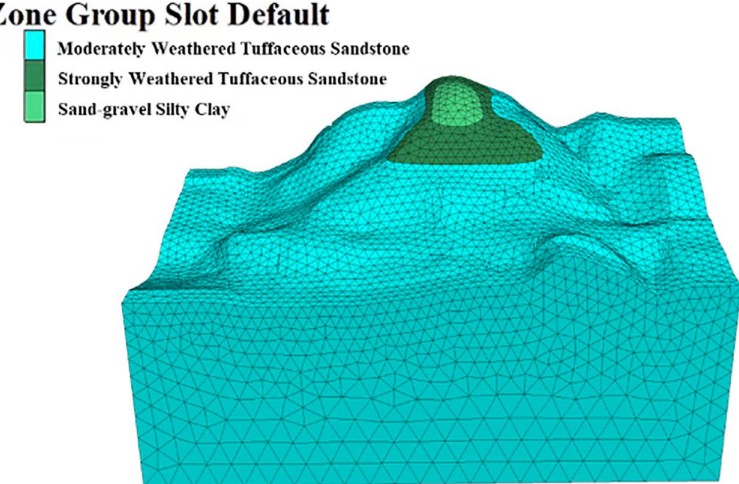

**Fig 11. 3D numerical model of the slope.** Original model generated by the authors. Copyright held by the authors, released under the Creative Commons Attribution License (CC BY 4.0).

**Table 2. Numerical analysis of slope rock calculates mechanical parameters.**

| Rock Layer | Density (kg/m³) | Bulk Modulus K (Pa) | Shear Modulus G (Pa) | Cohesion C (Pa) | Friction Angle (°) | Tensile Strength $\sigma_t$ (MPa) |
|---|---|---|---|---|---|---|
| Moderately Weathered Tuffaceous Sandstone | 2650 | 5.41E+09 | 2.64E+09 | 6.27E+06 | 62.18 | 3 |
| Strongly Weathered Tuffaceous Sandstone | 2250 | 1.56E+09 | 0.76E+09 | 5.3E+06 | 53.98 | 2 |
| Sand-gravel Silty Clay | 1860 | 0.023E+09 | 0.01E+09 | 1E+06 | 30 | 0.2 |

The slope has well-developed joints, and the structural surface statistics were obtained using the sampling window method, proposed by Kulatilake and Wu (1984) [26,27]. This method does not require measuring the trace length of structural surfaces but instead uses structural surface counting to determine the average trace length. A rectangular area with length a and width b is selected on the outcrop surface as the sampling window. The dimensions, orientation, group number, dip angle (α), and dip (β) of the sampled structural surfaces are recorded, as well as the relationship between each trace line and the sampling window, as shown in Fig 12.

The statistical data for the structural surfaces, obtained using the sampling window method in the field, are shown in Table 3. The bedding plane orientation is 332° (dip Angle) ∠ 24° (dip). Due to the limitations of the sampling window range, it was difficult to calculate the average radius and normal density of the bedding surface accurately, so appropriate corrections were made [28,29].

## 5.2. Invocation of the SMRM constitutive model

In this study, the natural slope calculation uses the Statistical Rock Mechanics (SMRM) constitutive model. The SMRM model is based on the stress-strain relationship in rock mass elasticity presented in Equation 1 in Chapter 4, which describes the elastic-plastic deformation behavior of the rock mass. Compared to traditional elastic models, SMRM considers the heterogeneity, fractures, and other characteristics within the rock mass, as well as the mechanical properties of the rock and structural surfaces, providing a more accurate representation of the complex mechanical characteristics of the rock mass.

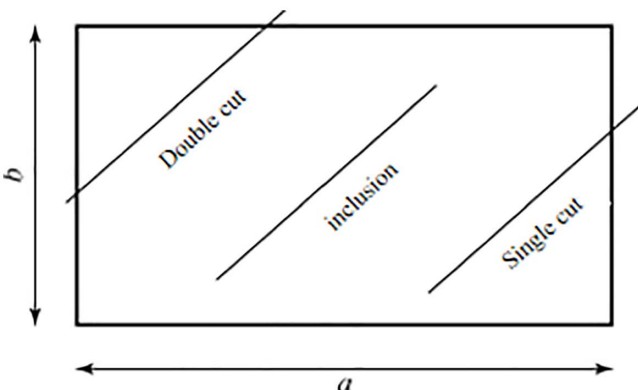

**Fig 12. Relationship between sampling window and fracture cleavage.**

**Table 3. Statistical window structure plane parameter table.**

| Group | Dip Angle (°) | Dip (°) | Average Radius (m) | Normal Density (1/m) | Cohesion C (Pa) | Friction Angle (°) |
|---|---|---|---|---|---|---|
| 1 | 340 | 24 | 10 | 6 | 0 | 30 |
| 2 | 40 | 67 | 1.99 | 0.45 | 0 | 30 |
| 3 | 45 | 72 | 3.16 | 0.41 | 0 | 30 |

Due to the strict requirements of numerical calculations for plasticity models, the previous SMRM constitutive module did not address the issue of calculating plastic deformation in the rock mass. However, since the stress-strain relationship in statistical rock mechanics itself includes elastic-plastic deformation, the SMRM constitutive module was updated to include elastic-plastic deformation calculations. Furthermore, in order to meet engineering needs, the module also includes extended calculations for commonly used indicators in geotechnical engineering.

Based on the custom constitutive interface provided by FLAC3D software, the SMRM constitutive model and parameter extension calculations were written into header files (ModelSMRM.h) and calculation modules (ModelSMRM.cpp) in C++ language and compiled into a FLAC3D 6.0 callable file (ModelSMRM.dll). On top of the traditional numerical analysis for calculating stress and displacement, the module adds extended calculations for commonly used geotechnical engineering indicators, including rock mass deformation modulus, SMRM rock mass quality score, failure probability of rock mass elements, and stability coefficients of rock mass elements. Based on these extended calculations, value contour maps for each indicator can be output. This module leverages the advantages of numerical computation in displaying spatial patterns, showing the spatial variation characteristics of these indicators, which facilitates grasping macroscopic trends in engineering analysis.

## 5.3. Application of the SMRM model in FLAC3D

### 5.3.1. SMRM constitutive anisotropy analysis.
Fig 13 shows the displacement contour calculated using the Mohr-Coulomb (MC) model, while Fig 14 shows the simulation based on the SMRM model. Except for differences in the structural surface parameters, the other model parameters and calculation conditions remain consistent, and the analysis profile was chosen from the same location on the slope face to ensure that the comparison results accurately reflect the impact of structural surface parameters on displacement field distribution characteristics. The total displacement magnitudes in Figs 15 and 13 were extracted from the centroids of all tetrahedral elements in FLAC3D and then mapped

**Fig 13. M-C Displacement Contour.**

into nephograms using volumetric averaging. Displacement in both models develops after gravity loading; specifically, the deformation shown in Fig 14 (SMRM model) is dominated by shear slip along the gently dipping structural planes (24°).

The displacement cloud map based on the MC constitutive model shows that the displacement distribution is smooth, with no significant local displacement anomalies or concentrated displacement. This distribution characteristic suggests that the MC model assumes isotropic material behavior, resulting in uniform and regular deformation.

In contrast, the SMRM constitutive model introduces the effects of structural surfaces, significantly altering the displacement field distribution, making it more complex. The displacement cloud map shows that the blue areas exhibit noticeable displacement concentration, contrasting with the bottom areas. The concentration zone extends along a specific direction, showing typical anisotropic features. This concentration phenomenon arises from local slip or shear effects caused by the structural surface settings, and the direction of the concentration zone aligns closely with the direction of the structural surface. This indicates that the SMRM model accurately represents the anisotropic properties of the material.

Measurement of the concentrated band in the SMRM displacement cloud map shows that the measured direction is 23°, which shows excellent agreement with the 24° dip angle of the bedding orientation. This bedding plane orientation was determined through field sampling window measurements, as documented in Table 3. The 1° deviation may stem from grid division accuracy in the numerical calculations, nonlinear material deformation characteristics, or the influence of boundary conditions and stress concentration effects. Nevertheless, this deviation falls within a reasonable range for complex geological models or numerical simulations, and the direction of the concentration zone remains highly consistent with the structural surface properties, demonstrating that the SMRM model can accurately reflect the anisotropic behavior of the material.

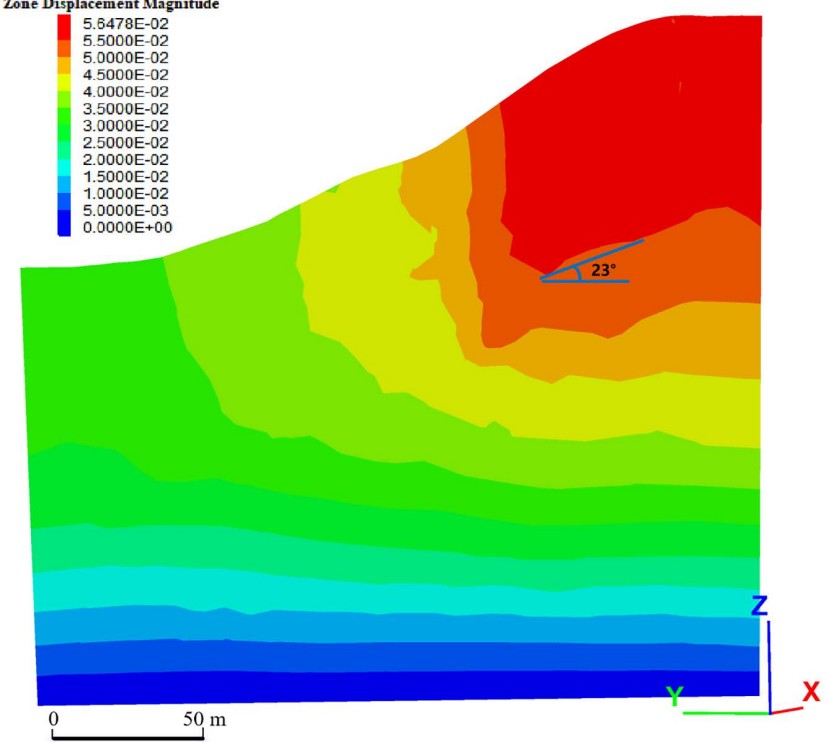

**Fig 14. SMRM Displacement Contour.**

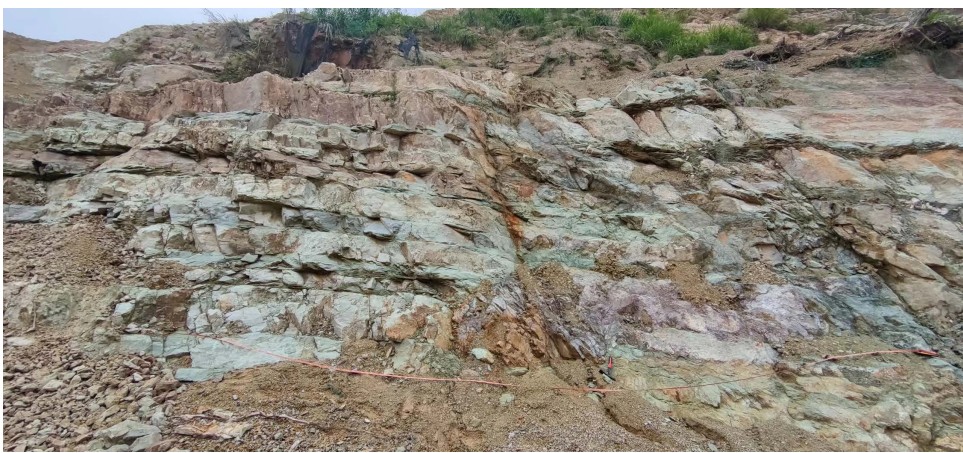

**Fig 15. The area to be measured in the field.** (Original photograph taken by the authors during fieldwork at the Kezhu Expressway. Copyright held by the authors, released under the Creative Commons Attribution License (CC BY 4.0).).

The SMRM constitutive model, by incorporating the directional influence of structural surfaces, can more realistically reflect the material's mechanical behavior, especially in cases where weak or sliding surfaces undergo shear failure in specific directions, significantly exhibiting characteristics of concentrated displacement and directional deformation.

Therefore, the SMRM model is suitable for describing the deformation and failure behavior of materials with weak or structural surfaces, particularly in landslide or slope stability analysis, where it can more accurately predict slip directions and potential instability zones.

The MC constitutive model assumes that materials are homogeneous and isotropic, making it suitable for overall analysis without obvious structural surfaces or weak planes. However, it may have limitations in capturing the detailed features or local deformation behavior of materials, especially in accurately representing complex deformation characteristics caused by structural surfaces.

**5.3.2. Application of the SMRM Model in FLAC3D.** Using the ModelSMRM fractured rock constitutive model, the joint geometric parameters, mechanical parameters, and rock mechanics parameters were applied to the strata, and a static analysis was conducted for slope stability.

Fig 16 shows the deformation pattern of the slope, where the maximum displacement at the top of the slope is 5.79 cm. Fig 6 shows the principal stress cloud map of the slope (Note: the principal stress is positive for tension and negative for compression; stress units: Pa). From the Fig., it can be seen that the stress exhibits a layered distribution, with compressive stress proportional to depth. In the maximum principal stress of the slope, the initial stress field is nearly at zero pressure, which is favorable for slope stability. The slope surface is mainly under compression, with a small area of tensile stress concentration; however, the distribution range is small and does not significantly affect the overall stability of the slope. The minimum principal stress is dominated by compressive stress, with a minimum compressive stress of 0.0126 MPa on the slope surface, and no tensile stress region appears.

The elastic modulus cloud map of the slope is shown in Fig 7, where the minimum value of the elastic modulus, 0.074 GPa, is concentrated at the top of the slope, which also contains the base cover layer with lower strength. The rock mass quality cloud map of the slope is shown in Fig 8. Referring to the SMRM quality grading empirical relationship in Table 1, the base cover layer has a rock mass quality score of 15–20, corresponding to Grade IV rock mass quality. The overall surface rock mass quality score is 25–40, corresponding to Grade IV rock mass quality, and the bedrock quality score is 40–50, corresponding to Grade III rock mass quality. The SMRM rock mass quality cloud map provides more information about the spatial variation of rock mass quality, indicating that fixed rock mass quality grades are not suitable for real-world conditions.

The cloud maps of rock mass reinforcement demand, failure probability, and stability coefficient distribution are shown in Figs 6−8 and 11−16. The reinforcement demand reflects the need for additional support under the current conditions of the rock mass. Some areas require high-intensity support to enhance the stability of the rock mass. These areas usually correspond to regions with low quality grades, high failure probabilities, and low stability coefficients. As shown in Fig 9, in the area with the lowest quality grade, the reinforcement demand is highest, but it has not yet reached the point where reinforcement is needed. The maximum failure probability in this area is 60%, and the minimum stability coefficient is 1. This small area is insufficient to impact slope stability. This area also has the highest reinforcement demand, while the overall stability coefficient of the slope is around 2.0.

In the field of engineering rock mass, due to changes in geometric and mechanical boundary conditions, stress fields generally exhibit uneven distribution, leading to variations in rock mass quality depending on spatial location. In particular, areas with highly concentrated stress may cause rock burst phenomena, while the appearance of secondary tensile stress could lead to rock relaxation, reducing rock mass quality and weakening overall slope stability. Therefore, regions with poor rock mass stability should be the focus of engineering attention. However, these critical areas are generally limited in distribution and have clear boundaries. Based on this, handling the entire or large areas of rock mass comprehensively is inefficient and may lead to resource wastage. Through SMRM-based FLAC3D simulation analysis, weak areas in the rock mass can be accurately identified, targeted strategies can be developed, and resources can be concentrated on strengthening key areas, optimizing resource allocation, and maximizing engineering benefits Fig 17.

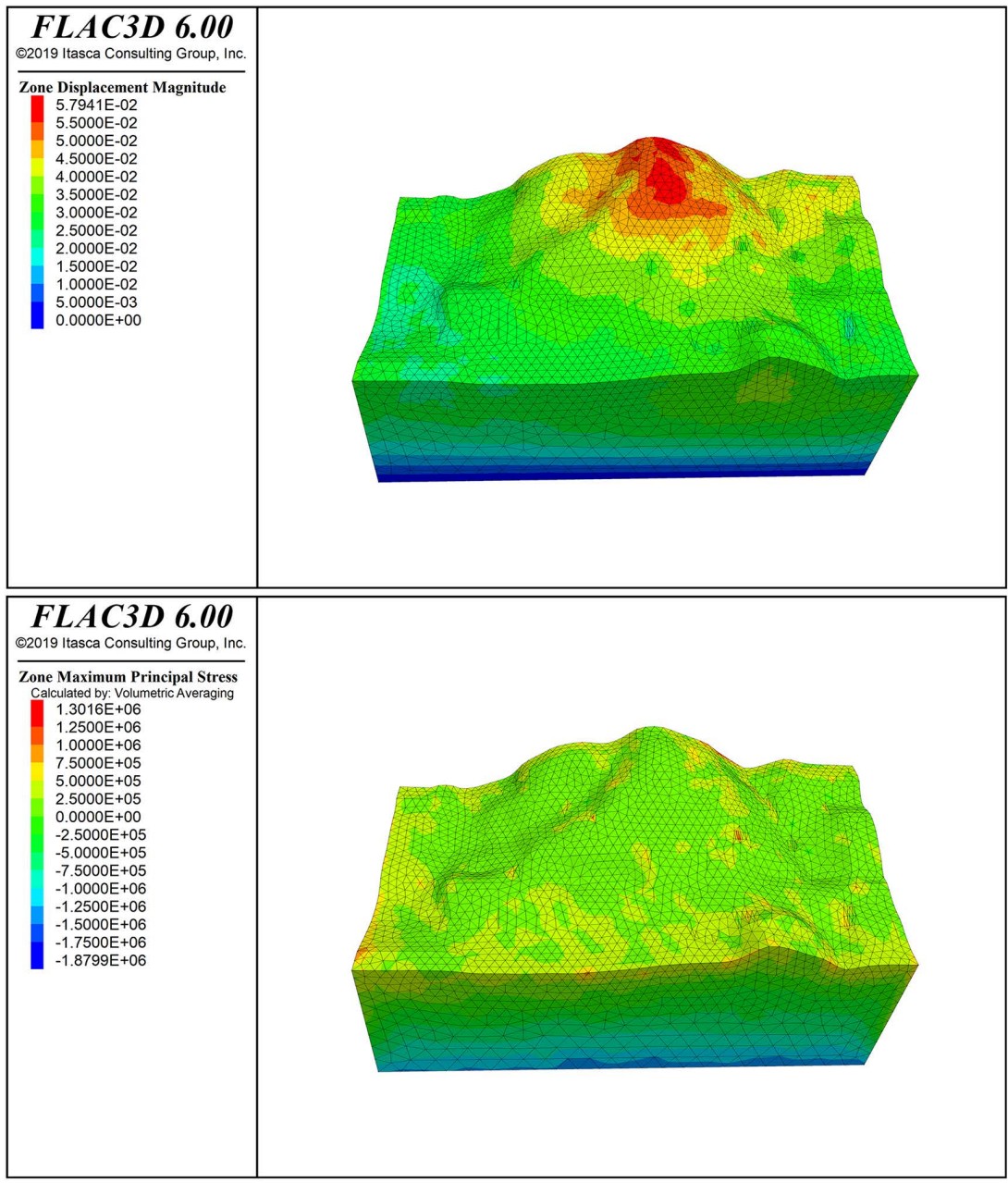

**Fig 16. Displacement Magnitude contour. (a)** Maximum principal stress contour. **(b)** Minimum principal stress contour.

## 6. Conclusion

This study establishes a UAV-assisted 3D modeling framework integrated with the SMRM constitutive model to address stability challenges in layered rock slopes. The main conclusions of the study are as follows:

(1) The "CC-Griddle-Flac3D" modeling method is proposed, which converts drone point cloud data into high-precision 3D numerical models, enabling rapid modeling under complex geological conditions.

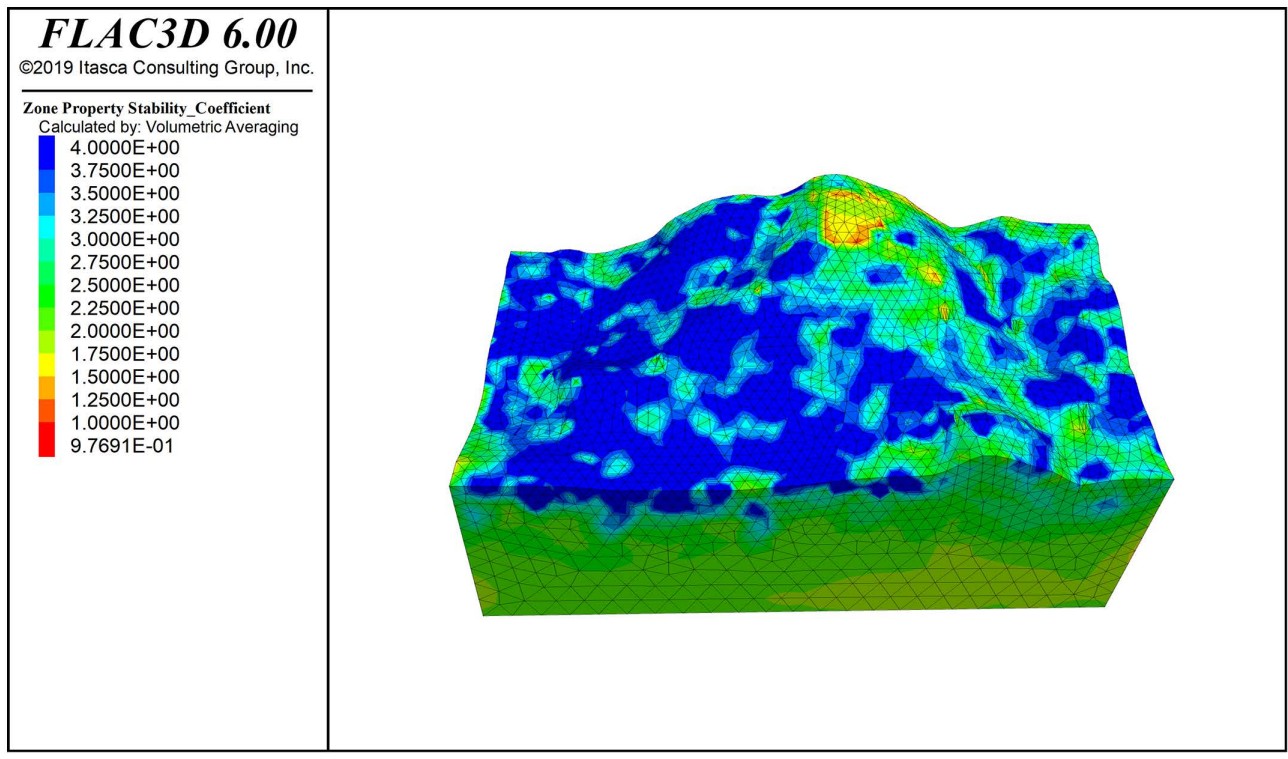

**Fig 17. stabitity_Coefficient contour.**

(2) Based on the elastic stress-strain relationship in statistical rock mechanics, this module extends traditional numerical analysis (such as stress and displacement calculations) by adding commonly used indicators in geotechnical engineering. These include the rock mass deformation modulus, SMRM rock mass quality score, failure probability of rock mass elements, and stability coefficients of rock mass elements.

(3) Comparative analysis shows that the SMRM constitutive model has significant advantages in simulating the anisotropic deformation of materials with structural surfaces. Despite the presence of minor numerical errors, their impact on the overall results is minimal, and the overall trend of the model is highly consistent with actual characteristics. Therefore, the SMRM constitutive model has high reliability and application value in deformation analysis and failure prediction under complex geological conditions.

(4) The SMRM constitutive model and its extended calculation module developed on the FLAC3D platform achieve the organic integration of rock mass parameter states and parameter fields. Through engineering case analysis, there is a good correspondence between the areas with the lowest stability coefficients and those with the maximum displacement, reinforcement demand, and failure probability. This allows for the precise identification of weak areas in the rock mass and the development of targeted strategies.

(5) The SMRM constitutive model is suitable for the actual rock mass with complex structural plane network, which can effectively analyze the influence of structural plane parameters on the mechanical properties of rock mass, especially for the jointed rock mass with multiple sets of structural planes, and can evaluate the rock mass engineering problems under the action of various structural planes. However, for highly fractured rock mass or soil slope, it may be

necessary to combine other constitutive models (such as strain softening model) for multi-constitutive coupling analysis to describe its mechanical behavior more comprehensively. In the future, the coupling method between SMRM and other constitutive models can be further studied to expand its applicability.

## Supporting information

**S1 File. Information about rock material parameters for slope stability analysis.** 10.6084/m9.figshare.29377223. (XLSX)

**S2 File. Information about Statistical window structure plane parameter table.** DOI:10.6084/m9.figshare.29377235. (XLSX)

## Acknowledgments

The authors are grateful to anonymous reviewers for their valuable suggestions.

## Author contributions

**Formal analysis:** Zhongliang Wang.

**Project administration:** Huagang Shan, Weiguo Wang, Jianyang Lu, Zhenghua Li.

**Software:** Jie Wu.

**Supervision:** Faquan Wu.

**Visualization:** Jie Wu.

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
