## [Decision Letter · Decision Letter 0]

28 May 2025

PONE-D-25-24746Integrated UAV Photogrammetry and SMRM Constitutive Modeling for Spatial Stability Assessment of Layered Rock SlopesPLOS ONE

Dear Dr. Wu,

Thank you for submitting your manuscript to PLOS ONE. After careful consideration, we feel that it has merit but does not fully meet PLOS ONE’s publication criteria as it currently stands. Therefore, we invite you to submit a revised version of the manuscript that addresses the points raised during the review process.

We look forward to receiving your revised manuscript.

Kind regards,

Zhengzheng Cao

Academic Editor

PLOS ONE

“This work was funded by Research Project of Zhejiang Provincial Department of Transport (No. 2024-GCKY-01).”

“Scientific Research Project of Zhejiang Provincial Department of Transportation ;2024GCKY01;.

The Central Guidance on Local Science and Technology Development Fund of Projects (2024ZY01041).”

6. We note that your Data Availability Statement is currently as follows: [All relevant data are within the manuscript and its Supporting Information files.]

7. When completing the data availability statement of the submission form, you indicated that you will make your data available on acceptance. We strongly recommend all authors decide on a data sharing plan before acceptance, as the process can be lengthy and hold up publication timelines. Please note that, though access restrictions are acceptable now, your entire data will need to be made freely accessible if your manuscript is accepted for publication. This policy applies to all data except where public deposition would breach compliance with the protocol approved by your research ethics board. If you are unable to adhere to our open data policy, please kindly revise your statement to explain your reasoning and we will seek the editor's input on an exemption. Please be assured that, once you have provided your new statement, the assessment of your exemption will not hold up the peer review process.

8. We note that Figures 2, 4 and 7 in your submission contain copyrighted images. All PLOS content is published under the Creative Commons Attribution License (CC BY 4.0), which means that the manuscript, images, and Supporting Information files will be freely available online, and any third party is permitted to access, download, copy, distribute, and use these materials in any way, even commercially, with proper attribution. For more information, see our copyright guidelines: http://journals.plos.org/plosone/s/licenses-and-copyright.

1. You may seek permission from the original copyright holder of Figures 2, 4 and 7to publish the content specifically under the CC BY 4.0 license.

9. We note that Figures 1 and 3 in your submission contain [map/satellite] images which may be copyrighted. All PLOS content is published under the Creative Commons Attribution License (CC BY 4.0), which means that the manuscript, images, and Supporting Information files will be freely available online, and any third party is permitted to access, download, copy, distribute, and use these materials in any way, even commercially, with proper attribution. For these reasons, we cannot publish previously copyrighted maps or satellite images created using proprietary data, such as Google software (Google Maps, Street View, and Earth). For more information, see our copyright guidelines: http://journals.plos.org/plosone/s/licenses-and-copyright.

1. You may seek permission from the original copyright holder of Figures 1 and 3 to publish the content specifically under the CC BY 4.0 license.

Reviewers' comments:

Reviewer's Responses to Questions

**Comments to the Author**

1. Is the manuscript technically sound, and do the data support the conclusions?

Reviewer #1: Yes

Reviewer #2: Yes

Reviewer #3: Yes

2. Has the statistical analysis been performed appropriately and rigorously? 

Reviewer #1: Yes

Reviewer #2: Yes

Reviewer #3: Yes

3. Have the authors made all data underlying the findings in their manuscript fully available?

Reviewer #1: Yes

Reviewer #2: Yes

Reviewer #3: Yes

4. Is the manuscript presented in an intelligible fashion and written in standard English?

Reviewer #1: Yes

Reviewer #2: Yes

Reviewer #3: Yes

5. Review Comments to the Author

Reviewer #1: 1.The existing charts have the problem of missing annotations, which may affect the readability and rigor of the data. As shown in Figure 8 (MC model displacement cloud diagram) and Figure 9 (SMRM model displacement cloud diagram), a scale and coordinate axis labels need to be added to clearly define the range of displacement values.

2. In Section 5.3, the study only mentioned the consistency between the simulation results and the inclination Angle of the structural plane, but lacked the support of measured data. It is necessary to strengthen the argumentation logic of "model validity". It is suggested that the author supplement the comparison between the on-site monitoring data and the numerical simulation results (such as the difference between the measured values and the simulated values of the displacement monitoring points) to verify the reliability of the model.

3.Provide a brief overview of the relevant background information. Some papers can be beneficial in the introduction. Development rule of ground fissure and mine ground pressure in shallow burial and thin bedrock mining area. Disaster-causing mechanism of spalling rock burst based on folding catastrophe model in coal mine

4. In Section 4.2, the current comparative analysis is not in-depth enough, and readers may have difficulty understanding the innovation of the SMRM model and its unique advantages in slope stability analysis. It is suggested that the author further explain the core differences between the SMRM model and traditional constitutive models (such as Mohr-Coulomb), especially how to quantify the anisotropy of rock mass through statistical mechanics theory.

5. Some references are not formatted properly. For instance, in reference 21, "841051" requires the author to explain the specific meaning of "841051".

Reviewer #2: 1. It is recommended to provide a detailed explanation of the theoretical foundation of the "SMRM constitutive model," including the definition of key parameters and its differences from conventional constitutive models, to enhance the rationale and innovation of model selection.

2. In the UAV point cloud data processing section, it is suggested to supplement information on acquisition accuracy (such as resolution, flight altitude, and overlap ratio) and error control methods to further clarify the impact of modeling quality on subsequent numerical analysis.

3. While the CC-Griddle-Flac3D coupled modeling process is efficient, it is advisable to provide a detailed presentation of the mesh partitioning strategy, element types, boundary condition settings, and other technical details to enhance model reproducibility.

4. It is recommended to further explain the calculation methods and physical significance of various rock mass indicators (such as Δσ₃ and failure probability) in the SMRM model, and systematically compare them with field observations.

5. The abstract of the article needs to be revised, and the importance of the article should be highlighted. The background and mechanism are not introduced clearly. Mechanical behavior and fracture mechanism of high-temperature granite cooled with liquid nitrogen for geothermal reservoir applications. Physics of Fluids, Brittleness evaluation of gas-bearing coal based on statistical damage constitution model and energy evolution mechanism. Journal of Central South University

6. In the conclusion section, it is recommended to further emphasize the applicability boundaries and limitations of the method, explaining whether it is suitable for different types of rock masses and various structure-controlled slope engineering conditions, while also proposing future optimization directions.

Reviewer #3: The study presents a integrated framework combining UAV photogrammetry, 3D numerical modeling (FLAC3D), and the SMRM constitutive model for layered rock slope stability analysis. While the methodological workflow is validated, the discussion would benefit from a brief comparative analysis with alternative approaches to contextualize its advantages.

1. How do the SMRM-based results compare to those from discrete element methods or finite eleent methods in handling structural discontinuities?

2. Could machine learning-assisted approaches offer complementary insights to the purely physics-based SMRM model?

3. While displacement alignment is promising, how might the accuracy/efficiency trade-offs compare to traditional inclinometer or LiDAR-based monitoring?

6. PLOS authors have the option to publish the peer review history of their article (what does this mean? ). If published, this will include your full peer review and any attached files.

**Do you want your identity to be public for this peer review?** For information about this choice, including consent withdrawal, please see our Privacy Policy .

Reviewer #1: No

Reviewer #2: No

Reviewer #3: No

---

## [Author Response · Author response to Decision Letter 1]

4 Oct 2025

Response to Editor

Dear Editor

Thank you very much for your email concerning our manuscript entitled " Integrated UAV Photogrammetry and SMRM Constitutive Modeling for Spatial Stability Assessment of Layered Rock Slopes” (ID: PONE-D-25-24746). We have carefully considered the points raised by the reviewers, and have made the following responses and changes. We deem PLOS ONE as a very excellent journal and hope the submitted revision can be accepted by this international journal.

Here, we would like to thank the reviewers for their insightful comments, and very much appreciate your patience, time and efforts in handling the reviewing process.

Best regards,

All authors

Journal requirements

Requirements 1: “ Please ensure that your manuscript meets PLOS ONE's style requirements, including those for file naming. The PLOS ONE style templates can be found at”

Authors' responses:

Thank you for your guidance. We have revised the manuscript to fully comply with requirements.

Requirements 2: “ Please note that PLOS ONE has specific guidelines on code sharing for submissions in which author-generated code underpins the findings in the manuscript. In these cases, we expect all author-generated code to be made available without restrictions upon publication of the work. ”

Authors' responses:

Thank the reviewers for their attention. We have clearly stated in the manuscript that the data used in this study and the code developed to produce the results can be obtained by contacting the corresponding author at a reasonable request, in accordance with the specific requirements of the journal for the code and data availability statement.

Requirements 3: “ In your Methods section, please provide additional information regarding the permits you obtained for the work. Please ensure you have included the full name of the authority that approved the field site access and, if no permits were required, a brief statement explaining why. ”

Authors' responses:

Thank the reviewers for their attention, because this study does not require a permit, because it does not involve the following :

• Humans (live or tissue), including studies that are observational, survey-based, or include any personal data

• Animals (live or tissue), including observational studies

• Cell lines that are not commercially available

• Field sampling

• Potential biosafety implications

Requirements 4: “ We note that the grant information you provided in the ‘Funding Information’ and ‘Financial Disclosure’ sections do not match.When you resubmit, please ensure that you provide the correct grant numbers for the awards you received for your study in the ‘Funding Information’ section. ”

Authors' responses:

Thanks to the reviewer, I have revised.

Requirements 5: “Thank you for stating the following in the Acknowledgments Section of your manuscript:

“This work was funded by Research Project of Zhejiang Provincial Department of Transport (No. 2024-GCKY-01).”We note that you have provided funding information that is currently declared in your Funding Statement. However, funding information should not appear in the Acknowledgments section or other areas of your manuscript. We will only publish funding information present in the Funding Statement section of the online submission form.

“Scientific Research Project of Zhejiang Provincial Department of Transportation ;2024GCKY01;.The Central Guidance on Local Science and Technology Development Fund of Projects (2024ZY01041).”

Please include your amended statements within your cover letter; we will change the online submission form on your behalf. ”

Authors' responses:

Thanks to the reviewer, I have revised.

Requirements 6: “ We note that your Data Availability Statement is currently as follows: [All relevant data are within the manuscript and its Supporting Information files.]

Please confirm at this time whether or not your submission contains all raw data required to replicate the results of your study. Authors must share the “minimal data set” for their submission. PLOS defines the minimal data set to consist of the data required to replicate all study findings reported in the article, as well as related metadata and methods (https://journals.plos.org/plosone/s/data-availability#loc-minimal-data-set-definition).”

Authors' responses:

Thank you for your professional guidance throughout the review process. We sincerely appreciate the time and expertise you dedicated to evaluating our work.

Data sharing at DOI : 10.6084 / m9.figshare.29377235 and DOI : 10.6084 / m9.figshare.29377223.

Requirements 7: “ When completing the data availability statement of the submission form, you indicated that you will make your data available on acceptance. We strongly recommend all authors decide on a data sharing plan before acceptance, as the process can be lengthy and hold up publication timelines. Please note that, though access restrictions are acceptable now, your entire data will need to be made freely accessible if your manuscript is accepted for publication. This policy applies to all data except where public deposition would breach compliance with the protocol approved by your research ethics board. If you are unable to adhere to our open data policy, please kindly revise your statement to explain your reasoning and we will seek the editor's input on an exemption. Please be assured that, once you have provided your new statement, the assessment of your exemption will not hold up the peer review process.”

Authors' responses:

Thank you for your guidance.We have stored the data in the public repository.

Requirements 8: “ We note that Figures 2, 4 and 7 in your submission contain copyrighted images. All PLOS content is published under the Creative Commons Attribution License (CC BY 4.0), which means that the manuscript, images, and Supporting Information files will be freely available online, and any third party is permitted to access, download, copy, distribute, and use these materials in any way, even commercially, with proper attribution. For more information, see our copyright guidelines: http://journals.plos.org/plosone/s/licenses-and-copyright.”

Authors' responses:

Thank you for your note regarding Figures 2, 4, and 7. We confirm these figures are original content created by the authors and do not contain any third-party copyrighted material:

Figures 2 and 7: Original photographs taken by author during field investigations at The Kezhu Expressway.

Figure 4: Original flowchart created by the authors to visualize the point cloud modeling workflow using CloudCompare、Rhino and Flac3d.

These figures are fully compliant with the CC BY 4.0 license.

Requirements 9: “ We note that Figures 1 and 3 in your submission contain [map/satellite] images which may be copyrighted. All PLOS content is published under the Creative Commons Attribution License (CC BY 4.0), which means that the manuscript, images, and Supporting Information files will be freely available online, and any third party is permitted to access, download, copy, distribute, and use these materials in any way, even commercially, with proper attribution. For these reasons, we cannot publish previously copyrighted maps or satellite images created using proprietary data, such as Google software (Google Maps, Street View, and Earth). For more information, see our copyright guidelines: http://journals.plos.org/plosone/s/licenses-and-copyright. ”

Authors' responses:

We sincerely appreciate your insightful suggestions regarding image copyright compliance. Your expertise was instrumental in enhancing the quality of our visual materials.

We have solved the copyright issues of Figure 1 as follows : Original Figure 1, because there may be copyright restrictions from commercial map providers. New Figure 1 : Generated by Natural Earth ( public domain ) : http://www.naturalearthdata.com/.

Figures 3 : Original photographs taken by author during field investigations at The Kezhu Expressway.This figures are fully compliant with the CC BY 4.0 license.

Response to Reviewer

1. Response to Reviewer #1

Comment 1: “The existing charts have the problem of missing annotations, which may affect the readability and rigor of the data. As shown in Figure 8 (MC model displacement cloud diagram) and Figure 9 (SMRM model displacement cloud diagram), a scale and coordinate axis labels need to be added to clearly define the range of displacement values.”

Authors' responses:

We sincerely thank the reviewers for their careful review and valuable suggestions. As suggested, we have carefully revised Figures 8 and 9 to improve their clarity and completeness. The modifications include:

Adding scale bars to explicitly indicate the displacement range.

Labeling the coordinate axes to clarify the spatial reference.

Ensuring the colorbar is clearly annotated with units and values.

Fig.8 M-C Displacement Contour Fig.9 SMRM Displacement Contour

Comment 2: “In Section 5.3, the study only mentioned the consistency between the simulation results and the inclination Angle of the structural plane, but lacked the support of measured data. It is necessary to strengthen the argumentation logic of "model validity". It is suggested that the author supplement the comparison between the on-site monitoring data and the numerical simulation results (such as the difference between the measured values and the simulated values of the displacement monitoring points) to verify the reliability of the model.”

Authors' responses:

We appreciate the reviewers constructive suggestion regarding the validation of the model’s reliability. In the revised manuscript,The revised sections have been highlighted in red for clarity, we have strengthened the argument by:

The displacement direction angle measured from the simulated SMRM displacement cloud map (23°) was compared with the bedding plane inclination angle of 24° obtained from field sampling window measurements (as shown in Table 3). The mere 1° deviation between them indicates that the model can accurately capture the anisotropic behavior of the material.

While direct on-site displacement monitoring data would further strengthen the validation, our current analysis demonstrates a strong correlation between the simulated displacement pattern and the structural inclination direction.

“Measurement of the concentrated band in the SMRM displacement cloud map shows that the measured direction is 23°, which shows excellent agreement with the 24° dip angle of the bedding orientation. This bedding plane orientation was determined through field sampling window measurements, as documented in Table 3.“

It is hoped that these changes will respond to your concerns and make our research more scientifically valuable and practical.

Thank you again for your careful review and guidance.

Comment 3: “Provide a brief overview of the relevant background information. Some papers can be beneficial in the introduction. Development rule of ground fissure and mine ground pressure in shallow burial and thin bedrock mining area. Disaster-causing mechanism of spalling rock burst based on folding catastrophe model in coal mine”

Authors' responses:

Thank you for your suggestion. I've reviewed the recommended literature :

(1) Development rule of ground fissure and mine ground pressure in shallow burial and thin bedrock mining area (DOI: 10.1038/s41598-024-77324-7). This article uses numerical simulation to explore the behavior of the mine overburden during the mining process, which complements the exploration of the use of numerical simulation in mine.

(2) Disaster-causing mechanism of spalling rock burst based on folding catastrophe model in coal mine (DOI: 10.1007/s00603-025-04497-6). This study performs Cracked Coal Panel stability study based on Folding Catastrophe model, which is similar to our stability study using the SMRM eigenstructure.

We've decided to incorporate these references into the introduction section of our manuscript. This addition will strengthen our literature review and provide a more comprehensive context for our study.

Comment 4: “In Section 4.2, the current comparative analysis is not in-depth enough, and readers may have difficulty understanding the innovation of the SMRM model and its unique advantages in slope stability analysis. It is suggested that the author further explain the core differences between the SMRM model and traditional constitutive models (such as Mohr-Coulomb), especially how to quantify the anisotropy of rock mass through statistical mechanics theory.”

Authors' responses:

Thank you for your valuable feedback. We sincerely apologize for the inconvenience caused by the absence of specific reference information in the revised manuscript.

The SMRM constitutive model combines the geometric probability model of rock mass structure, the energy principle of fracture mechanics and the continuum mechanics method. According to the energy superposition principle and the weakest ring theory, the constitutive model of jointed rock mass is established by considering the mechanical response of rock mass and structural plane. In the SMRM constitutive relation formula 2 in section 4.2, are the cosine of the structural surface 's normal direction reflects the anisotropy of the structural plane. In this paper, the ' 4.3 SMRM constitutive advantage ' is added to further explain the SMRM model and the traditional constitutive model. Below, highlight the revised part in red to show clarity.

4.3 SMRM constitutive advantage

Based on the theory of statistical rock mechanics ( SMRM ), the SMRM constitutive model developed by Flac3D platform is developed and improved in this study, which effectively enhances its application ability in numerical simulation of rock mass engineering. Compared with the traditional model, the constitutive advantages of SMRM are as follows :

( 1 ) The existing numerical calculation models of rock mass engineering are mainly based on continuous and isotropic media, and there are some limitations in the mechanical analysis of anisotropic media. Based on the geometric probability of rock mass structure and the energy superposition principle of fracture mechanics, the SMRM constitutive model superimposes the strain energy of rock mass under external force with the strain energy of structural plane network, and establishes a set of analysis theory for describing the mechanical behavior of equivalent continuous medium of rock mass.

( 2 ) Based on the advantages of Flac3D in spatial distribution, the SMRM constitutive model can directly show the spatial variation characteristics of rock mass related indexes, and provide scientific basis for grasping the macroscopic law in engineering analysis. The anisotropy of rock mass parameters can be characterized by means of stereographic projection and anisotropy coefficient. The non-uniformity can be intuitively reflected by the point cloud map of the spatial position of the specific engineering rock mass. For engineering problems, the stress field distribution of the model can be obtained by numerical calculation of the SMRM model, and based on this, the spatial cloud diagram of key parameters including rock mass elastic modulus, SMRM rock mass quality index, rock mass unit failure probability, rock mass reinforcement demand and stability coefficient can be generated. As shown in Figs.12-16.

( 3 ) The layered structural plane in the SMRM constitutive model is established in an 'implicit ' way. In the modeling process, the bedding structural plane is not considered first, and the corresponding bedding plane stress distribution can be automatically generated in the calculation by inputting the inclination and inclination of the bedding plane through the command flow. This method simplifies the modeling process and improves the modeling efficiency.

Comment 5: “Some references are not formatted properly. For instance, in reference 21, "841051" requires the author to explain the specific meaning of "841051".”

Authors' responses:

Thank you for your careful review. You are absolutely right that "84105

---

## [Decision Letter · Decision Letter 1]

16 Nov 2025

PONE-D-25-24746R1Integrated UAV Photogrammetry and SMRM Constitutive Modeling for Spatial Stability Assessment of Layered Rock SlopesPLOS ONE

Dear Dr. Wu,

Thank you for submitting your manuscript to PLOS ONE. After careful consideration, we feel that it has merit but does not fully meet PLOS ONE’s publication criteria as it currently stands. Therefore, we invite you to submit a revised version of the manuscript that addresses the points raised during the review process.

**The manuscript must be corrected in all points indicated by the reviewers, such as:**

1) Fig.1b/Fig.2C: The image resolution is insufficient.

2) Numerical calculations are based on the actual slope topography. Since the area has dense vegetation, please specify whether vegetation removal was performed during point cloud processing.

3) The modeling process description in Section 3 is overly detailed. It is recommended to use a flowchart for illustration.

4) The meaning of Em1111 in Formula 3 is unclear.

5) The grouping in Figures 4 and 5 lacks annotations.

6) In Table 1, the unit for tensile strength is incorrect.

7) Line 372: The expression "The bedding orientation is 332°∠24°" is unclear.

8) For Table 2, please clarify whether the structural surfaces are simulated using standard contact models. If so, provide specific parameters and value determination methods.

9) Regarding the displacement nephograms in Figures 8 and 9, please specify: a) What processing they are based on. b) The cause of displacement generation.

We look forward to receiving your revised manuscript.

Kind regards,

Claudionor Ribeiro da Silva

Academic Editor

PLOS ONE

**Journal Requirements:**

Reviewers' comments:

Reviewer's Responses to Questions

**Comments to the Author**

1. If the authors have adequately addressed your comments raised in a previous round of review and you feel that this manuscript is now acceptable for publication, you may indicate that here to bypass the “Comments to the Author” section, enter your conflict of interest statement in the “Confidential to Editor” section, and submit your "Accept" recommendation.

Reviewer #1: All comments have been addressed

Reviewer #4: All comments have been addressed

2. Is the manuscript technically sound, and do the data support the conclusions?

Reviewer #1: Yes

Reviewer #4: Yes

3. Has the statistical analysis been performed appropriately and rigorously? 

Reviewer #1: Yes

Reviewer #4: N/A

4. Have the authors made all data underlying the findings in their manuscript fully available?

Reviewer #1: Yes

Reviewer #4: Yes

5. Is the manuscript presented in an intelligible fashion and written in standard English?

Reviewer #1: Yes

Reviewer #4: Yes

6. Review Comments to the Author

**Reviewer #1:** This method enhances the spatial visualization of rock mass parameters, enabling targeted reinforcement strategies. The study provides a validated technical pathway for rapid slope stability evaluation in geologically complex regions, supporting data-driven disaster prevention decisions.

**Reviewer #4:**  Fig.1b/Fig.2C: The image resolution is insufficient.

Numerical calculations are based on the actual slope topography. Since the area has dense vegetation, please specify whether vegetation removal was performed during point cloud processing.

The modeling process description in Section 3 is overly detailed. It is recommended to use a flowchart for illustration.

The meaning of Em1111 in Formula 3 is unclear.

The grouping in Figures 4 and 5 lacks annotations.

In Table 1, the unit for tensile strength is incorrect.

Line 372: The expression "The bedding orientation is 332°∠24°" is unclear.

For Table 2, please clarify whether the structural surfaces are simulated using standard contact models. If so, provide specific parameters and value determination methods.

Regarding the displacement nephograms in Figures 8 and 9, please specify:

a) What processing they are based on.

b) The cause of displacement generation.

7. PLOS authors have the option to publish the peer review history of their article (what does this mean? ). If published, this will include your full peer review and any attached files.

**Do you want your identity to be public for this peer review?** For information about this choice, including consent withdrawal, please see our Privacy Policy .

Reviewer #1: No

Reviewer #4: No

---

## [Author Response · Author response to Decision Letter 2]

17 Dec 2025

Dear Editor

Thank you very much for your email concerning our manuscript entitled " Integrated UAV Photogrammetry and SMRM Constitutive Modeling for Spatial Stability Assessment of Layered Rock Slopes” (ID: PONE-D-25-24746). We have carefully considered the points raised by the reviewers, and have made the following responses and changes. We deem PLOS ONE as a very excellent journal and hope the submitted revision can be accepted by this international journal.

Here, we would like to thank the reviewers for their insightful comments, and very much appreciate your patience, time and efforts in handling the reviewing process.

Best regards,

All authors

Issues 1:“Fig.1b/Fig.2C: The image resolution is insufficient.”

Authors' responses:

We appreciate the reviewer's attention to detail and agree that improving the image resolution is necessary for clarity. Enclosed are the high-resolution versions of Figures 1b and 2C.

Fig. 1 Geographic location of the study area

Fig.2 Images of the study area: (c) Engineering geological cross-section

Issues 2:“Numerical calculations are based on the actual slope topography. Since the area has dense vegetation, please specify whether vegetation removal was performed during point cloud processing.”

Authors' responses:

Thank you for your valuable comment. In response to your inquiry, we have clarified in the revised manuscript (lines 152-155) that vegetation removal was performed during point cloud processing.

“The Cloth Simulation Filter (CSF) method was used to remove vegetation data and retain ground information. This process separated vegetation points from ground points, ensuring that the numerical calculations considered only terrain features.”

Issues 3:“The modeling process description in Section 3 is overly detailed. It is recommended to use a flowchart for illustration.”

Authors' responses:

Thank you for your valuable feedback. We agree that the modeling process description in Section 3 was too detailed, and we have revised the section to provide a more concise overview. We have simplified the text and added a flowchart(Fig. 3) to illustrate the key steps in the modeling process more clearly. The updated Section 3 now focuses on the essential steps and provides an easier-to-follow representation of the methodology.

Issues 4:“The meaning of Em1111 in Formula 3 is unclear.”

Authors' responses:

Thank you for pointing this out. We have revised Formula 3 by changing Em1111 to Em, and we have added a clarification to the text: "where Em is the elastic modulus of the rock mass."

Issues 5:“The grouping in Figures 4 and 5 lacks annotations.”

Authors' responses:

Figure 4 already contains subplot labels; Figure 5 now has additional annotations for each rock layer.

Fig.6 3D numerical model of the slope

Issues 6:“In Table 1, the unit for tensile strength is incorrect.”

Authors' responses:

The unit in column 4 of Table 1 has been changed from “Pa” to “MPa”.

Issues 7:“Line 372: The expression "The bedding orientation is 332°∠24°" is unclear.”

Authors' responses:

We agree that the single-angle notation may be unfamiliar to some readers.

The sentence (now line 321) has been rewritten as:

“The bedding plane orientation is 332° (dip Angle) ∠ 24° (dip). ”

Issues 8:“For Table 2, please clarify whether the structural surfaces are simulated using standard contact models. If so, provide specific parameters and value determination methods.”

Authors' responses:

No, a user-defined constitutive model (SMRM model) was developed to simulate the mechanical behavior of structural surfaces in this study. The detailed formulation and validation of the SMRM model are presented in Section 4.

The geometric parameters of the structural surfaces (e.g., dip angle, dip direction, trace length, and normal density) were obtained from field measurements using the sampling window method.

The strength parameters (cohesion C and friction angle φ) were empirically determined based on engineering experience and typical values for moderately weathered tuffaceous sandstone, as shown in Table 2.

Issues 9:“Regarding the displacement nephograms in Figures 8 and 9, please specify: a) What processing they are based on. b) The cause of displacement generation.”

Authors' responses:

a)The following sentence has been added in Section 5.3.1 (lines 353–355):

The total displacement magnitudes in Figures 8 and 9 were extracted from the centroids of all tetrahedral elements in FLAC3D and then mapped into nephograms using volumetric averaging.

b)The subsequent sentence (lines355–357) now reads:

“Displacement in both models develops after gravity loading; specifically, the deformation shown in Fig. 10 (SMRM model) is dominated by shear slip along the gently dipping structural planes (24°).”

---

## [Decision Letter · Decision Letter 2]

4 Jan 2026

Integrated UAV Photogrammetry and SMRM Constitutive Modeling for Spatial Stability Assessment of Layered Rock Slopes

PONE-D-25-24746R2

Dear Dr. Wu,

We’re pleased to inform you that your manuscript has been judged scientifically suitable for publication and will be formally accepted for publication once it meets all outstanding technical requirements.

Kind regards,

Claudionor Ribeiro da Silva

Academic Editor

PLOS One

Additional Editor Comments (optional):

Reviewers' comments:

Reviewer's Responses to Questions

**Comments to the Author**

1. If the authors have adequately addressed your comments raised in a previous round of review and you feel that this manuscript is now acceptable for publication, you may indicate that here to bypass the “Comments to the Author” section, enter your conflict of interest statement in the “Confidential to Editor” section, and submit your "Accept" recommendation.

Reviewer #1: All comments have been addressed

Reviewer #4: All comments have been addressed

2. Is the manuscript technically sound, and do the data support the conclusions?

Reviewer #1: Yes

Reviewer #4: Yes

3. Has the statistical analysis been performed appropriately and rigorously? 

Reviewer #1: Yes

Reviewer #4: Yes

4. Have the authors made all data underlying the findings in their manuscript fully available?

Reviewer #1: Yes

Reviewer #4: Yes

5. Is the manuscript presented in an intelligible fashion and written in standard English?

Reviewer #1: Yes

Reviewer #4: Yes

6. Review Comments to the Author

Reviewer #1: Layered rock slopes with structural discontinuities pose significant challenges for stability assessment under complex geological conditions. This study presents an integrated approach combining unmanned aerial vehicle (UAV) photogrammetry, 3D numerical modeling, and the Statistical Mechanics of Rock Masses (SMRM) constitutive model to evaluate slope stability.

It can be accepted.

Reviewer #4: (No Response)

7. PLOS authors have the option to publish the peer review history of their article (what does this mean? ). If published, this will include your full peer review and any attached files.

**Do you want your identity to be public for this peer review?** For information about this choice, including consent withdrawal, please see our Privacy Policy .

Reviewer #1: No

Reviewer #4: No

---

## [Editor Report · Acceptance letter]

PONE-D-25-24746R2

PLOS One

Dear Dr. Wu,

I'm pleased to inform you that your manuscript has been deemed suitable for publication in PLOS One. Congratulations! Your manuscript is now being handed over to our production team.

Kind regards,

on behalf of

Dr. Claudionor Ribeiro da Silva

Academic Editor

PLOS One